# Primate anterior insular cortex represents economic decision variables proposed by prospect theory

You-Ping Yang [1,2,4], Xinjian Li[2,3,4] & Veit Stuphorn [1,2,3 ✉]

In humans, risk attitude is highly context-dependent, varying with wealth levels or for different potential outcomes, such as gains or losses. These behavioral effects have been modelled using prospect theory, with the key assumption that humans represent the value of each available option asymmetrically as a gain or loss relative to a reference point. It remains unknown how these computations are implemented at the neuronal level. Here we show that macaques, like humans, change their risk attitude across wealth levels and gain/loss contexts using a token gambling task. Neurons in the anterior insular cortex (AIC) encode the 'reference point' (i.e., the current wealth level of the monkey) and reflect 'loss aversion' (i.e., option value signals are more sensitive to change in the loss than in the gain context) as postulated by prospect theory. In addition, changes in the activity of a subgroup of AIC neurons correlate with the inter-trial fluctuations in choice and risk attitude. Taken together, we show that the primate AIC in risky decision-making may be involved in monitoring contextual information used to guide the animal's willingness to accept risk.

[1] Department of Psychological and Brain Sciences, Johns Hopkins University, 3400N. Charles St., Baltimore, MD 21218-2685, USA. [2] Zanvyl Krieger Mind/Brain Institute, 3400N. Charles St., Baltimore, MD 21218-2685, USA. [3] Department of Neuroscience, Johns Hopkins University School of Medicine, 3400N. Charles St., Baltimore, MD 21218-2685, USA. [4]These authors contributed equally: You-Ping Yang, Xinjian Li. ✉email: veit@jhu.edu

Uncertainty about the possible outcomes of chosen actions is a basic feature of all human and animal decision making. How our nervous system deals with this uncertainty is therefore a fundamental question in cognitive neuroscience. Decisions under uncertainty depend on an individual's risk attitude, i.e., the willingness to accept uncertainty about the outcome (risk) in exchange for possibly better outcomes than a safer alternative. Risk attitude is strongly influenced by context. Humans show different risk attitudes when facing risky gains versus risky losses[1]. The abundance of economic resources in the environment and the current wealth of subjects also modulate an individual's risk attitude[2–5]. Prospect theory[6], the most influential[7] and wide-ranging[8] descriptive model of decision-making under risk, explains these context-dependent changes in risk attitude using two critical concepts about the cognitive processes underlying value estimation. First, prospect theory assumes that humans evaluate possible future outcomes either as gains or as losses relative to a reference point (i.e., the current wealth, resources, or state of the subject). Second, human's sensitivities to changes in value are different for losses and gains. Specifically, humans are more sensitive to changes in value for losses as compared to gains (i.e., losses loom larger than gains). Thus, humans' choices can be manipulated by framing an identical outcome as either a gain or loss using verbal instructions, and by varying the current wealth of subjects that change the point of reference. Despite the success of prospect theory as a descriptive model of risky choices, it remains unclear if its underlying assumptions match the framework for value estimation implemented on the neuronal level.

Human imaging experiments and lesion studies have identified a network of brain areas that are active during decision-making under risk[9–14]. Of particular interest is the anterior insular cortex (AIC), a large heterogeneous cortex in the depth of the Sylvian fissure. Human fMRI studies have suggested a crucial role of AIC in representing subjects' current internal states[15,16], and in risk-aversive behavior[12,13]. Lesions in the AIC have also been documented to affect the risk-attitude of human patients[17,18]. Moreover, recording studies in monkeys have shown that AIC neurons encode reward expectation[19,20]. Based on these findings, we hypothesized that the AIC neurons may encode behaviorally relevant value information in the framework suggested by prospect theory. AIC would represent the current state of the subject (the reference point) as well as reference-dependent value signals that differ in loss or gain context (asymmetrical value functions in loss and gain). Together, these representations in AIC would influence a subject's risk attitude in decision making.

To test this hypothesis, we developed a token-based gambling task and recorded single neuron activities from the AIC of two macaque monkeys engaged in this task. We first examined whether and how monkeys changed their risk attitude in various behavioral contexts. Next, we identified AIC neurons representing factors that influence risk attitude, such as starting token number, gain or loss outcome, and uncertainty. Finally, we determined whether the AIC neurons encoding these factors also predict the monkey's choice or risk attitude.

Here, we show that monkeys, like humans, have different risk attitudes depending on the gain/loss context, and that AIC neurons encode reference-dependent value signals, consistent with the asymmetric value function as postulated by the Prospect theory. In addition, both the monkeys' choices and the activity of AIC neurons are strongly influenced by the number of tokens that the monkeys possessed at the start of the trial, indicating that the momentary wealth level served as a reference point. Intertrial fluctuations in the activity of AIC neurons encoding these variables were correlated with the monkeys' choices and risk attitude. Taken together, these results support our hypothesis that the primate AIC encodes the reference point and reference-dependent value signals, and that these value representations of available options modulate the animal's willingness to accept risk in the current behavioral context.

## Results

Two monkeys were trained in a token-based gambling task (Fig. 1a). In this task, the monkey had to collect a sufficient number of tokens (≥6) to receive a standard fluid reward (600 μl water). Because the maximum number of tokens that could be earned in a single trial was three, the monkeys had to accumulate the necessary tokens over multiple trials (Supplementary Fig. 1). On each choice trial, the monkey chose between a gamble option (uncertain outcome) and a sure option (certain outcome), which could result in gaining or losing tokens. The number of tokens to be won or lost was indicated by the color of the target cues, while the probability was indicated by the relative proportion of each colored area (Fig. 1b). To investigate whether the monkeys' risk attitude was different for gains and losses, we presented either only gain or only loss options on any given trial. Thus, in the gain context, the monkey had to choose between a sure option that resulted in a certain token increase, whose size varied across trials, versus a gamble option that could result in a large increase or no increase at all, with varying outcome probabilities across trials (Fig. 1b, left). In the loss context, the monkey had to choose between a certain loss and an uncertain option that could result in no loss at all or a large loss (Fig. 1b, right). The monkeys selected the chosen option by making a saccade to the corresponding target cue. After a short delay (450–550 ms), the outcome was revealed, and the number of currently owned tokens (token assets) was updated. If a trial ended with a token number less than 6 (e.g., 4), these tokens (e.g., 4) were kept as the start tokens for the next trial. If a trial ended with a token number larger than 6 (e.g., 8), water was delivered and the remaining tokens (e.g., 8−6 = 2) were rolled over to the start of the next trial.

Both monkeys learned the task, as indicated by the observation that their fixation behavior was strongly influenced by their token assets. Monkeys fixated faster (Supplementary Fig. 2a–c) and were less likely to break their fixation (resulting in abortion of the trial) (Supplementary Fig. 2d–f) when they had larger token assets at the start of the trial, and when they received more tokens from the previous trial. These results suggest that monkeys understood the use of tokens as secondary reinforcers, and thus were more motivated when they owned more and received more tokens, before they actually earned the primary reinforcer (the fluid reward).

**Monkeys' risky choices are influenced by gain/loss context and current token assets.** We found that monkeys' choices were influenced by the gain/loss context. Both monkeys were more likely to choose the gamble option than the sure option (Fig. 1c; one-sided $t$-test; Monkey G, P(Gamble) = 59%, $p < 10^{-4}$; Monkey O, P(Gamble) = 67%, $p < 10^{-4}$) and were even more likely to do so in the gain context than in the loss context (Fig. 1c; one-sided paired $t$-test, $p < 10^{-4}$ for both Monkey G and Monkey O). We have also found that monkeys' choices were influenced by the number of tokens they owned at the start of the trial (current token assets), but differently for gains and losses. In the gain context, the probability of the monkey choosing the gamble option (P(Gamble)) decreased as the token assets increased (Fig. 1d; green dashed line; regression analysis; Monkey G, $\beta = -0.044$, $p < 10^{-4}$; Monkey O, $\beta = -0.035$, $p < 10^{-4}$). In contrast, in the loss context P(Gamble) increased with increasing token assets (Fig. 1d; red dashed line; regression analysis; Monkey G, $\beta = 0.028$, $p < 10^{-4}$; Monkey O, $\beta = -0.001$, $p = 0.8$). Thus, as the monkeys

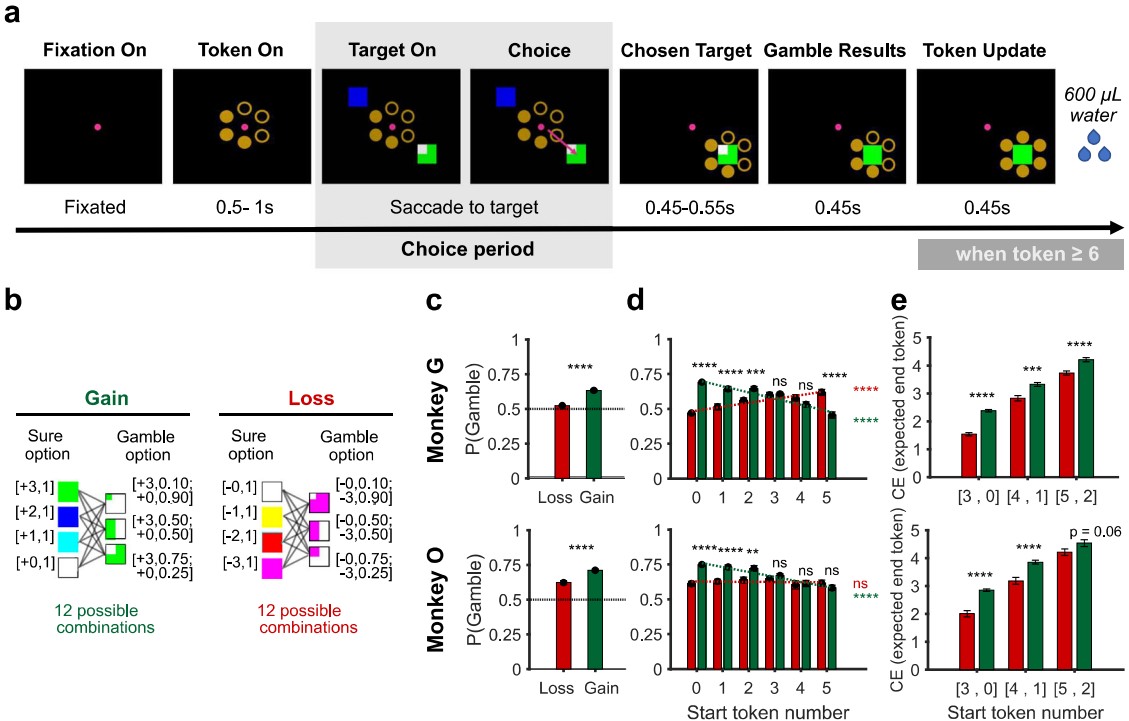

**Fig. 1 Behavioral performance of monkeys in the token-based gambling task. a** Task design. The monkey was informed about the currently owned token number (indicated by the filled dots surrounding the fixation spot) and chooses a sure or gamble option by making a saccade to the desired reward option. After the outcome was revealed, the token number was updated. The monkey was rewarded whenever it collected six tokens or more at the end of the trial. Shadowed area indicates the choice period, during which the neuronal activity was analyzed. **b** Reward option set. Each option was defined by the number of tokens gained or lost (indicated by color) and the probability of each outcome (indicated by the portion of colored area). The outcomes and probabilities are indicated in brackets next to each sure and gamble option. The trials were divided into gain and loss conditions. Each gamble option was paired against all sure options ranging from best ($+3$ for the gain and 0 for the loss context) to worst (0 for the gain and $-3$ for the loss context) possible gamble outcomes, resulting in 24 combinations. See "Methods" section for details. **c** The probability of monkey choosing the gamble option in gain and loss contexts across all sessions. One-sided paired $t$-test. **d** The probability of monkey choosing the gamble option as a function of gain/loss context and start token number. Pairwise comparison for each start token level: one-sided paired $t$-test; black statistical markers. Trend across start token level: regression; green and red statistical markers. **e** Subjective value of gamble-pairs, measured by their certainty equivalent (CE), that resulted in the same distribution of expected end tokens, but had different start token number and thus represented gain (green) or loss (red) of tokens. one-sided permutation test. **c–e** $n = 37$ sessions for both monkey G and monkey O. Data in the gain and loss context are colored in green and red, respectively. Data are presented as mean values ± SEM. ns not statistically significant (i.e., $p > 0.05$), **$p < 10^{-2}$, ***$p < 10^{-3}$, ****$p < 10^{-4}$.

owned more token assets, they became more risk-averse for further gains (i.e., less willing to gamble for a greater win), but were more risk-seeking for avoiding a potential loss. These results are in line with the observation of humans that human subjects tend to be more risk-aversive when facing a potential gain, and more risk-seeking when facing a potential loss as their own asset increases[1].

Monkeys' response time (RT, the interval between stimulus onset and the saccade initiation) was also influenced by these contextual factors. Both monkeys responded slower in the loss than in the gain context (Supplementary Fig. 3a, b; one-sided permutation test; monkey G: $RT_{gain} = 205$ ms, $RT_{loss} = 247$ ms, $p < 10^{-3}$; monkey O: $RT_{gain} = 175$ ms, $RT_{loss} = 206$ ms, $p < 10^{-3}$), and when they owned more tokens (Supplementary Fig. 3c, d; regression analysis; monkey G: $\beta_{StartTkn} = 2.83$, $p = 0.19$; monkey O: $\beta_{StartTkn} = 3.50$, $p < 10^{-2}$). This suggests that monkeys chose more carefully when facing a potential loss, and when they are getting closer to six tokens for the water reward. In addition, RT was also influenced by expected value of the chosen option, and the difference of expected value of gamble and sure option (Supplementary Fig. 3e–h).

Prospect theory implies the use of a relative value framework, where the value of an outcome depends on the change in assets relative to a reference point. Alternatively, the monkeys could use an absolute value framework, where the value of an outcome depends on the final asset number. To test, which of these value frameworks is used by the monkeys, we compared trials with the same outcome in terms of final token number, but which resulted from either gaining or losing tokens. For example, consider a trial with a start token number of 0, in which a gamble option with an equal probability of gaining 3 or 0 tokens is offered versus a sure option of gaining two tokens. The expected end token outcomes of this trial (owning 3 or 0 tokens, each with $p = 0.5$ versus owning two tokens with $p = 1$) are identical to a trial with a start token number of three, in which a gamble option with an equal probability of losing 3 or 0 tokens is offered versus a sure option of losing one token (Supplementary Fig. 4a). By systematically matching pairs of this type, we found three sets of gamble and sure options that reached the same final token number by either gaining or losing tokens (Supplementary Fig. 4b, c). We compared the subjective values (SVs) of these gambles across gains and losses using the model-free certainty equivalent method (Supplementary Fig. 4d)[21]. If monkeys used an absolute value framework, the SV of the gamble options should not be different for gain and loss trials. However, we found that the SV of gamble options strongly depended on whether the outcome represented a gain or a loss of tokens (Fig. 1e; one-sided permutation test; monkey G & O: $p < 10^{-4}$; see also Supplementary Fig. 4c). This strongly indicated that the monkeys used a relative, rather than an absolute, value framework to guide risky choices.

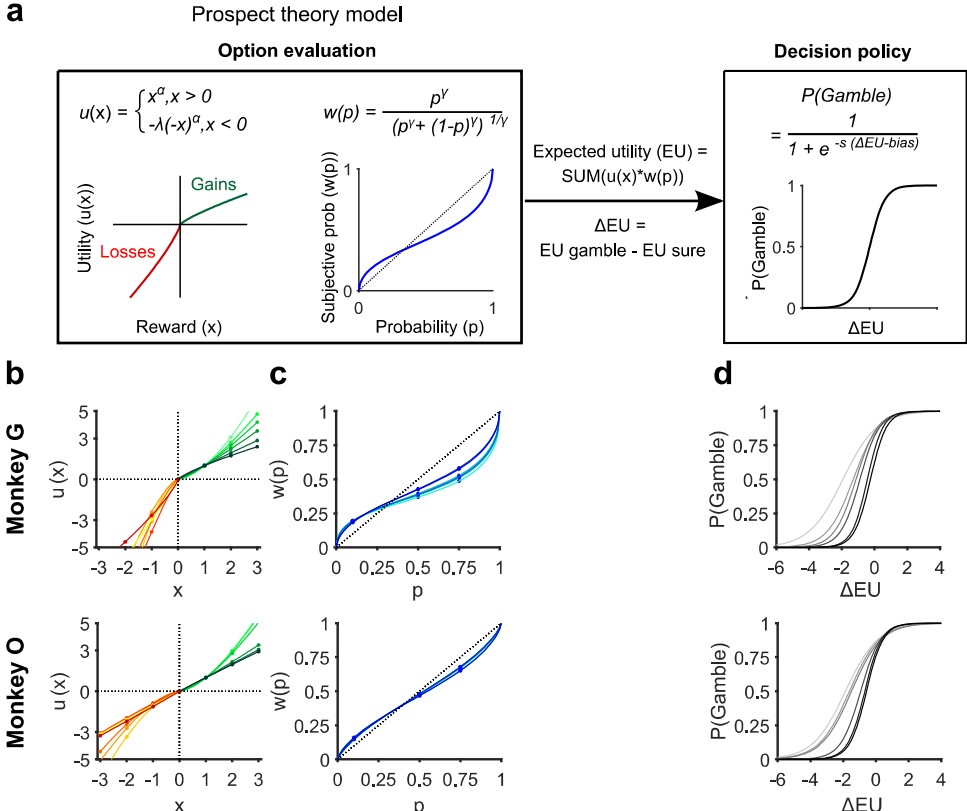

**Fig. 2 Behavioral modeling of the prospect theory model. a** Behavior modeling. The model consists of two parts: first in the process of option evaluation, the expected utility (EU) of each option was calculated as the product of a utility function and a probability weighting function. Both functions are nonlinear as per the Prospect theory (PT) hypothesized. The expected utility difference between the two options (ΔEU) was then used to determine the probability of choosing the gamble option via a logistic function -i.e., decision policy. **b, c** The best fit utility functions (**b**), probability weighting functions (**c**), and the decision policies (**d**), based on the observed performance. Color gradients represent results from trials with different start token numbers (light to dark: 0 to 5).

**Prospect theory model of risk-attitude adjustment.** After confirming that monkeys' choice behavior was influenced by core contextual factors critical for prospect theory (PT), we used this model to describe choice behavior. The key component of the PT model is to weight gains, losses, and probabilities differently before they are combined to form a subjective evaluation of the option (Fig. 2a). The relative gains and losses are mapped onto corresponding utility as follows: $u(x) = x^\alpha$ when $x > 0$ (reward outcome in gain) and $u(x) = -\lambda(-x)^\alpha$ when $x < 0$ (reward outcome in loss). The utility function component $\alpha$ captures risk-attitude. Convex utility (with $\alpha > 1$) indicates risk-seeking, while subjects are more sensitive to differences in larger rewards. Concave utility (with $\alpha < 1$) indicates risk-avoidance, due to diminishing marginal utility. The utility function component $\lambda$ captures loss-aversion, the idea that losses loom larger than equivalent gains. $\lambda > 1$ indicates more sensitivity to losses than gains and $\lambda < 1$ indicates more sensitivity to gains than losses.

To capture the influence of tokens on different components, we modeled behavior for each start token number independently. In the gain context, both monkeys were risk-seeking ($\alpha > 1$) when the start token number was low, but they became risk neutral or risk-averse when the start token number increased (Fig. 2b; light to dark green lines indicate increasing start token number). Estimated $\alpha$ was negatively modulated by the start token number (regression analysis; Monkey G, $\beta = -0.16$, $p < 10^{-4}$; Monkey O, $\beta = -0.14$, $p < 10^{-4}$). In monkey G, the utility functions were consistently steeper for losses than for gains (Fig. 2b; yellowish to red lines indicate increasing start token number; one-tailed $t$-test: $\lambda > 1$; Monkey G, $p < 10^{-4}$ for all start token numbers). Thus,

monkey G showed loss-aversion. However, in monkey O the utility functions were not consistently steeper for losses than for gains and $\lambda$ values varied around 1. This indicated that monkey O was equally sensitive to gains and losses and thus showed no loss aversion. There was no significant difference for the estimated $\lambda$ across different start token numbers for either monkey (regression analysis; Monkey G, $\beta = 0.03$, $p = 0.69$; Monkey O, $\beta = 0.01$, $p = 0.4$).

Objective probabilities are mapped onto a subjective weighting function as follows: $w(p) = p^\gamma/(p^\gamma + (1-p)^\gamma)^{1/\gamma}$ [2,6,22]. $\gamma > 1$ indicates an S-shape subjective probability mapping (overestimated for large probabilities and underestimated for small probabilities), $\gamma < 1$ indicates an inverse S-shape subjective probability mapping (underestimated for large probabilities and overestimated for small probabilities), and $\gamma = 1$ indicate a linear mapping of objective probabilities. Both monkeys showed an inverse S-shaped mapping of probabilities (Fig. 2c; one-sided $t$-test: $\gamma < 1$; both monkeys, $p < 10^{-4}$ for all start token numbers), confirming previous findings[22–24]. The mappings were slightly influenced by increasing start token numbers in one monkey (light blue to dark blue lines; regression analysis; Monkey G, $\beta = 0.02$, $p < 10^{-4}$), but not at all in the other one (Monkey O, $\beta = 0.0004$, $p = 0.89$).

After calculating the expected utility ($EU = u(x) * w(p)$) of each option based on prospect theory, we estimated the probability to choose the gamble option, P(Gamble), by passing expected utility difference between options (ΔEU) through a softmax function with parameters $s$ and bias: $P(\text{Gamble}) = \frac{1}{1+e^{-s(\Delta EU - \text{bias})}}$, where $s$ controls choice stochasticity

**Table 1 Model comparison.**

| Model | Start token number | DF | α | λ | γ | Inverse Temperature (s) | Directional bias to gamble (bias) | −2*LLmax | BIC |
|---|---|---|---|---|---|---|---|---|---|
| Prospect theory (PT) model | 0 | 5 | 1.63, 1.58 | 3.04, 1.12 | 0.52, 0.73 | 0.96, 1.01 | −1.93, −1.86 | 5100, 6494 | 5146, 6540 |
| | 1 | 5 | 1.43, 1.55 | 2.40, 0.93 | 0.56, 0.79 | 1.26, 1.09 | −1.30, −1.62 | 1522, 1481 | 1561, 1516 |
| | 2 | 5 | 1.30, 1.48 | 3.07, 0.87 | 0.55, 0.77 | 1.52, 1.09 | −1.13, −1.49 | 1418, 1379 | 1458, 1416 |
| | 3 | 5 | 1.15, 1.11 | 3.86, 0.91 | 0.55, 0.78 | 1.69, 1.63 | −0.89, −0.93 | 2403, 2427 | 2445, 2468 |
| | 4 | 5 | 0.95, 1.01 | 2.96, 1.03 | 0.63, 0.78 | 2.11, 1.93 | −0.44, −0.68 | 1072, 894 | 1110, 929 |
| | 5 | 5 | 0.79, 0.97 | 2.65, 1.12 | 0.62, 0.73 | 2.05, 1.96 | −0.25, −0.58 | 1347, 1178 | 1385, 1215 |
| Expected value (EV) model | 0 | 2 | — | — | — | 1.89, 1.70 | −0.67, −1.00 | 7204, 7121 | 7231, 7149 |
| | 1 | 2 | — | — | — | 2.01, 1.72 | −0.64, −0.98 | 1856, 1600 | 1879, 1623 |
| | 2 | 2 | — | — | — | 2.19, 1.58 | −0.61, −0.99 | 1746, 1455 | 1770, 1478 |
| | 3 | 2 | — | — | — | 2.12, 1.65 | −0.57, −0.90 | 2813, 2427 | 2838, 2499 |
| | 4 | 2 | — | — | — | 2.18, 1.86 | −0.31, −0.67 | 1219, 908 | 1242, 930 |
| | 5 | 2 | — | — | — | 1.63, 1.83 | −0.19, −0.57 | 1587, 1178 | 1609, 1232 |

The table summarizes for each model the likelihood maximizing (best) parameters average across sessions ($n = 37$ for both monkeys) and its fitting performances for each monkey.
*DF* degrees of freedom of the model, *α* parameter for utility curvature, *λ* parameter for loss-modulated utility curvature, *γ* parameter for probability weighting function, *LLmax* maximal log likelihood, *BIC* Bayesian Information Criterion.
Comparing the model fit of PT model and EV model: one-sided *t*-test; Monkey G, $p < 10^{-4}$, $p < 10^{-4}$, $p < 0.05$, $p < 0.05$, $p = 0.09$, and $p < 0.05$ for start token number 0–5, respectively; Monkey O, $p < 10^{-2}$, $p < 10^{-3}$ $p = 0.55$, $p = 0.62$, $p = 0.95$, and $p = 0.80$ for start token number 0–5, respectively.
Comparing the BIC of PT model and EV model: one-sided *t*-test; Monkey G, $p < 10^{-4}$, $p < 10^{-4}$, $p < 10^{-2}$, $p < 0.05$, $p < 0.05$, and $p < 0.01$; Monkey O, $p < 10^{-2}$, $p < 10^{-4}$, $p = 0.22$, $p = 0.17$, $p = 0.20$, and $p = 0.34$ for start token number 0–5, respectively.

and bias represents the tendency to choose the gamble option independent of the value calculation process. The monkeys showed a consistent tendency to choose the gamble option (Fig. 2d; leftward shift of the choice function in; one-sided *t*-test: both monkeys, $p < 10^{-4}$ for all start token numbers). This tendency decreased when the start token number increased (Fig. 2d; light gray to black lines; regression analysis; Monkey G, $\beta = 0.32$, $p < 10^{-4}$; Monkey O, $\beta = 0.28$, $p < 10^{-4}$), indicating monkeys became less risk-seeking as their wealth levels increased. Moreover, the choice functions for both monkeys became steeper, that is monkeys' choices became less stochastic, when the start token number increased (regression analysis; Monkey G, $\beta = 0.23$, $p < 10^{-4}$; Monkey O, $\beta = 0.22$, $p < 10^{-4}$). This result, combined with the token effect on response time, indicates that choices became slower but less stochastic when token assets increased, which suggests a speed-accuracy tradeoff.

In sum, the PT model describes the behavioral result well (Fig. 1c, d) and predicts the monkeys' choices better than the expected value model that does not assume nonlinearity in utility and probabilities (Supplementary Fig. 5). This was also true after considering the different number of free parameters (see quantitative model evaluation in Table 1). This suggests that the monkeys behavior was not simply aimed to maximize reward probability. Instead, key components of the prospect theory model are important for explaining the monkeys' behavior in our task.

**Anterior Insula neurons encode decision-related variables that influence risk-attitude.** To determine the neuronal basis underlying prospect theory, we recorded 240 neurons in the AIC of two macaque monkeys (monkey G: 142 neurons; monkey O: 98 neurons) working in the token gambling task. The recording locations are shown in Fig. 3a (more details in Supplementary Fig. 6). We analyzed the neuronal activity in the choice period (i.e., the time from target onset to saccade initiation) to determine if AIC neurons carried signals that could influence decision making. We based this analysis on activity during forced choice trials ($321 \pm 77$ trials per neuron), in which only one option was presented. In general, the AIC neurons showed weak spatial selectivity. Only 7% (17/240) of all AIC neurons showed a significant effect of spatial location on neuronal activity (one-way ANOVA, $p < 0.05$). We therefore ignored spatial target configuration for the remaining analysis.

The AIC neurons encoded three basic decision-related types of variables, representing value-related, token asset-related, and risk-related signals (Fig. 3 and Table 2). To quantitatively characterize the variables that each AIC neuron encodes during the choice period, we examined the activity of each neuron using a series of linear regression models (for details see "Methods" section). For each neuron, we identified the best fitting model using the Akaike information criterion and classified it into different functional categories according to the variables that were most likely encoded by the neuronal activity.

The majority of recorded AIC neuron activity (62%; 149/240) encoded at least one decision-related variable (task-related neurons: $p < 0.05$ for the coefficient of a specific variable in the best-fitting multiple linear regression model; Fig. 3b, more details in Table 2). Examples of these neuronal signals are shown in Fig. 3c–l (see Supplementary Fig. 7 for raster plots). The activity of the AIC neurons that carried value-related signals was correlated with the expected value of the options, using a relative value framework (i.e., token gains/losses). We found five basic types: neurons encoding (1) value across both gains and losses (Fig. 3c), (2) gain/loss category (Fig. 3d), behavioral salience (Fig. 3e), value only for losses (Fig. 3f), and value only for gains (Fig. 3g). The activity of the AIC neurons that carried token asset-related signals was correlated with the number of tokens owned at the beginning of the trial. We found three basic types: neurons encoding (1) token number in a parametric fashion (Fig. 3h), (2) high/low token level (Fig. 3i), and (3) preferred token number (Fig. 3j). The activity of the neurons carrying a risk-related signal was correlated with outcome variance. We found two basic types: neurons encoding risk (1) in a categorical and (2) in a parametric fashion (Fig. 3k). Finally, a small number of AIC neurons reflected the expected value of options in an absolute value framework (i.e., end token number) (Fig. 3l). A substantial number of AIC neurons (34%; 50/149) showed mixed selectivity and encoded more than one decision-related variable (Fig. 3b). The distributions of neural type classification were similar across the two monkeys (Supplementary Table 1).

The largest proportion of AIC neurons (70%; 105/149) reflected information about the currently owned token number. These token-encoding neurons used three different frameworks for encoding token assets. The first group (12%; 13/105) carried a Parametric token signal (Fig. 3h). These AIC neurons monotonically increased ($n = 11$) or decreased ($n = 2$) their activity

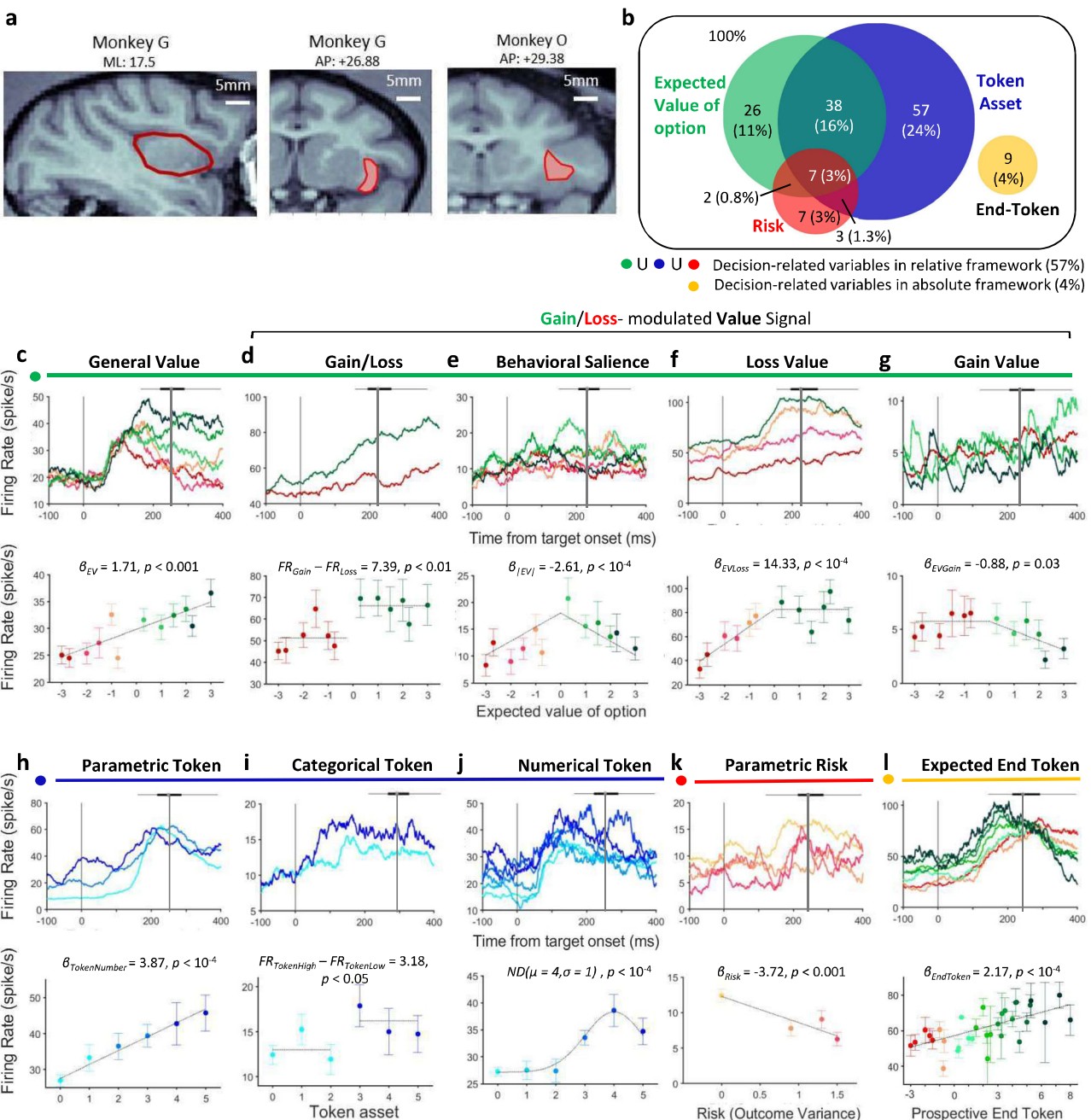

**Fig. 3 AIC neurons encode diverse task-related variables in forced-choice trials. a** MRI images showing the area of recording of each monkey. Left and middle: sagittal (left) and coronal (middle) view of the insular cortex of monkey G. Right: coronal view of the insular cortex of monkey O. **b** Venn diagram of the neurons encoding four types of task-related variables in the forced-choice trials. Green: expected value of option; Blue: start token number; Red: risk (variability of potential outcomes); Yellow: expected end token number. **c–g** Example neurons showing a variety of patterns by which gain/loss context and/or the expected option value (EV) were encoded. **c** General value signal: monotonic encoding of value across gain/loss contexts. **d** Gain/Loss value signal: categorical encoding of gain/loss context. **e** Behavioral salience signal: monotonic encoding of value in gain and loss context but with inverse directions. **f** Loss value signal: encoding of value only in the loss context. **g** Gain value signal: encoding of the value only in the gain context. **h–j** Example neurons showing a variety of patterns by which the token information was encoded. **h** Parametric token signal: monotonic encoding of the start token number. **i** Categorical token signal: categorical encoding of the start token number in low and high level. **j** Numerical token signal: neuronal response tunes to a specific number of start token (here 4). **k** Example neuron encoding parametric risk (i.e., outcome variance). **l** Example neuron encoding of the expected end token number. **c–l** Upper panels: spike density function (SDF), aligned on target onset ($t = 0$). Lower panels: mean firing rate of each example neuron at different levels of specific task-related variable. Mean firing rate (presented as mean values ± SEM) was calculated using the window from target onset to saccade initiation, which varied across trials. The saccade onset distribution is represented as a boxplot on top of each SDF. The box plot indicates median (vertical middle line), 25th, 75th percentile (box) and 5th and 95th percentile (whiskers). For clarity, when plotting the SDF, data were grouped together, as indicated by the color codes.

**Table 2 Summary of the number and percentage of significant responding neurons in different subsets of neuron types for all recorded AIC neurons.**

| Gain/Loss value | | | | | Token | | | Risk | | Expected End Token |
|---|---|---|---|---|---|---|---|---|---|---|
| Gain/Loss | Loss Value | Gain Value | Behavioral Salience | General Value | Parametric Token | Categorical Token | Numerical Token | Parametric Risk | Categorical Risk | |
| 13 (5%) 73 (30%) | 29 (12%) | 4 (2%) | 13 (5%) | 14 (6%) | 13 (5%) 105 (44%) | 11 (5%) | 81 (38%) | 9 (4%) 19 (8%) | 10 (4%) | 9 (4%) |

with the number of token assets. The second group (11%; 11/105) carried a Categorical token signal (Fig. 3i). These AIC neurons categorized all possible token numbers into a high [3, 4, 5] and a low [0, 1, 2] token level. Likely, this reflects a fundamental distinction between a token level, for which it is impossible that the monkey will earn reward at the end of the current trial (because the monkey can only earn a maximum of three tokens in one trial), and a high token level that makes it possible to earn a reward in the current trial. The third, and largest, group (77%; 81/105) carried a Numerical token signal. These AIC neurons are number-selective and are tuned for a preferred number (here four, example neuron in Fig. 3j). We used a Gaussian function to fit this activity pattern. The AIC neurons carrying a Numerical token signal covered the entire scale from 0 to 5 tokens with some neurons having each of the possible token amounts as their preferred number.

The second largest group of AIC neurons reflected information about the value of the options (49%; 73/149). A subset of this group of AIC neurons (19%; 14/73), carrying a General value signal (Fig. 3c), encoded the option values in a monotonically rising (18%; 13/73) or falling (1%; 1/73) fashion, for both gains and losses. Such neurons could either encode expected value or expected utility, but our data were not sufficient to distinguish between these possibilities. This kind of value signal is not gain/loss context sensitive. However, the neuronal activity of all other subsets of value-encoding neurons varied largely as a function of the way they represented value across the Gain and the Loss context (Fig. 3d–g). One group of these value-encoding neurons (18%; 13/73) carried a categorical Gain/Loss signal (Fig. 3d) that categorized each option as gain or loss, regardless of the expected value. In addition, we found AIC neurons (18%; 13/73) that represented value in both the gain and loss context, but with inverse correlations of neural activity and value (Fig. 3e). These neurons likely carried a Behavioral salience signal. Most interestingly, we found two other groups of AIC neurons carrying Loss value (Fig. 3f) or Gain value signals (Fig. 3g), respectively. These neurons represented a value signal, but only in either the loss or the gain context. We encountered more Loss value neurons (40%; 29/73) than Gain value neurons (6%; 4/73). The larger number of neurons encoding Loss value fits with human neuroimaging findings that suggest a role for the anterior insula in encoding aversive stimuli and situations[1,13].

Human neuroimaging data suggest that the anterior insular cortex encodes the riskiness of options[25,26]. Indeed, we found AIC neurons encoding risk-related signals, with risk defined as outcome variance (13%; 19/149). Half of these AIC neurons (47%; 9/19) encoded a Parametric risk signal (Fig. 3k) that encoded the risk of the various options continuously across both gains and losses. The other half of these AIC neurons (53%; 10/19) encoded a categorical risk signal that categorized options into safe or uncertain.

In the analysis so far, we have used a relative framework for value. Expected value was defined as token changes relative to a reference point (the start token number). However, value could also be defined in an absolute framework (i.e., the final token

number at the end of the trial). We tested for AIC neurons that represented expected absolute value, which is the expected end token number weighted by the probability of each outcome. However, we found only a very small number (6%; 9/149) carrying an End token signal (Fig. 3l).

The majority of AIC neurons showed activity pattern that matched several predictions of prospect theory. First, we found that many AIC neurons encode the wealth level of the monkey, i.e., the token number at the start of the trial. Within the context of our task, this variable represented the reference point relative to which the gain or loss options are evaluated. Simultaneously, this variable also indicates the current state of progress and indicates how close the monkey is to achieving the next reward. Second, many other AIC neurons reflect in their activity whether the offer is a gain or a loss. Some of them encoded the context, i.e., whether the options were presented in a gain or loss context. Other neurons represented a gain/loss-specific value signal in a parametric manner exclusively. Third, only very few neurons encode expected absolute value. Taken together, these three findings strongly imply that the primate AIC uses a relative value encoding framework, anchored to a reference point that reflects the current state of the monkey, as suggested by prospect theory.

**Value-encoding neurons in AIC exhibited contextual modulation predicted by the Prospect theory**. The majority of value-encoding AIC neurons were context-modulated (Fig. 3d–g). A strong assumption of Prospect theory is that changes in relative value are not encoded symmetrically across gains and losses. Indeed, the monkeys' behavior indicated that they were more sensitive to objective value differences in the loss than the gain context (i.e., steeper utility functions in the loss than that in gain context in Fig. 2b). We therefore investigated whether and how value signals across the AIC population showed matching differences in their sensitivity for gains and losses. We examined the absolute value of the standardized regression coefficients (SRC) of Loss-Value Neurons in the loss context and that of Gain-Value Neuron in the gain context. At the population level, we found indeed that Loss value signals and Gain Value signals had different sensitivities to changes in value. Specifically, the normalized $|SRC|$ of Loss-Value Neurons in the loss context were larger than that of Gain-Value Neuron in the gain context (Fig. 4a, one-sided permutation test; mean of $|SRC_{loss}|$ = 2.97, mean of $|SRC_{gain}|$ = 2.05, $p = 0.054$; unsigned SRC for losses and gains were indicated in red and green, respectively). This suggests that the AIC neurons encoding value signals were more sensitive to increasing loss than increasing gain (Fig. 4b).

Moreover, the sensitivity of value change in gain or loss context were also influenced by the wealth level. Normalized $|SRC|$ of Loss-Value Neurons in the loss context became smaller as the wealth level increased (Fig. 4c, left; one-sided permutation test; mean of $|SRC_{loss}|$ in low wealth level = 3.49, mean of $|SRC_{loss}|$ with high wealth Level= 2.47, $p = 0.017$; unsigned SRC in the loss context for low or high wealth levels were indicated in orange and red, respectively). Normalized $|SRC|$ of Gain-Value Neurons in the gain

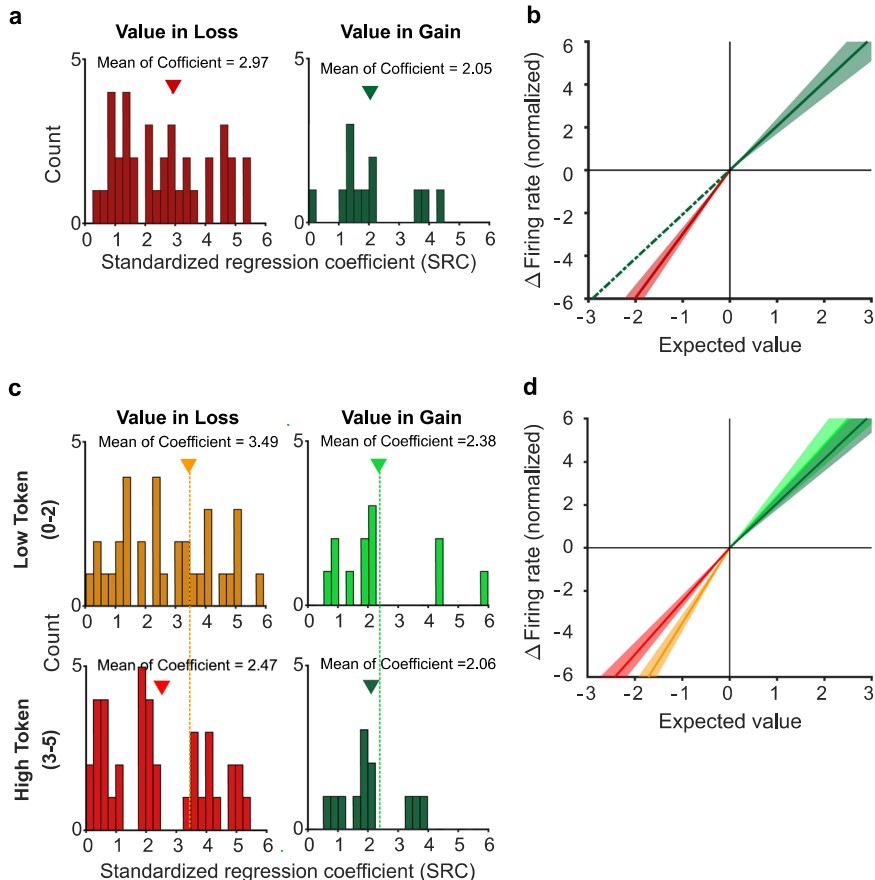

**Fig. 4 Gain-value and Loss-value neurons exhibit differential sensitivity to value change in the gain and loss context. a, b** Neuronal sensitivity to gain and loss estimated by linear regression of firing rates against expected option value of Loss-value neurons ($n = 39$) in loss-context trials; and for Gain-value neurons ($n = 12$) in gain-context trials. See "Methods" section for details of the neuron selection. **a** Distribution of the standardized regression coefficients (SRC). For cross-context comparison, the absolute value of SRCs (|SRCs|) were plotted. Each count represents one neuron. Inverted triangle: mean of the distribution. **b** The change in firing rate of Loss and Gain value neurons as a function in ΔEV (line: mean SRC value of the distributions shown in **a**; shaded region: ±SEM) in the loss-context and gain-context, respectively. Note that the slope of the red line is steeper than the slope of green line ($p = 0.053$, one-sided permutation test), indicating that Loss value neurons are more sensitive to EV change than Gain value neurons, mirroring the pattern observed from behavior (Fig. 2b). **c, d** Changes in neuronal sensitivity to gain and loss as a function of start token level was estimated by performing linear regression of firing rates against expected option value separately for low and high start token level. **c** Distribution of the |SRCs| of Loss Value neurons in loss-context and |SRCs| of Gain Value neurons in gain-context, split by start token levels. Each count represents one neuron. Inverted triangle: mean of the distribution. **d** Mean slope of regression based on the SRCs from **c**. Note that for both gain-contexts and loss-contexts, the slope becomes shallower as the token level increases, consistent to the pattern observed from behavior (Fig. 2b). **b, d** Loss value neurons and Gain Value neurons in loss-context and gain-context, are colored in red and green, respectively. Color gradients represent results from trials with different start token levels (light: low [0–2]; dark: high [3–5]). Data are presented as mean values ± SEM.

context also became smaller as the wealth level increased. However, this trend did not reach a significant level (Fig. 4c, right; one-sided permutation test; mean of $|\text{SRC}_{\text{gain}}|$ in low wealth level = 2.38, mean of $|\text{SRC}_{\text{gain}}|$ in high wealth level = 2.06, $p = 0.29$; unsigned SRC in gain context in low or high wealth levels were indicated in light and dark green, respectively). Again, this wealth level-sensitive effect on AIC value coding (Fig. 4d) is consistent with the fact that monkeys became less sensitive to objective value change when the wealth level increased (i.e., utility functions became flatter in both the loss and gain context when the wealth level increased; Fig. 2b).

**Choice-attitude and risk-attitude-related response modulations of AIC neurons.** So far, we have shown AIC neurons encoding decision-related variables in the forced-choice trials. Since the gain/loss-specific value and the token asset signals in AIC were present before the choice was made, these signals could be in a

position to influence the monkey's decisions. Therefore, we characterize next the relationship between all AIC neurons and behavioral choice, using the choice trials (369 ± 124 trials per neuron). To quantify how well the neuronal activity of an AIC neuron predicts the choice of the monkey, we computed an area under curve (AUC) value as a measure of the cell's discrimination ability, using receiver operation characteristic (ROC) analysis[27]. The AUC is a measure relating trial-to-trial fluctuation in neuronal activity to fluctuations in internal state (here implicit risk-attitude) or behavioral choice (here explicit choice of gamble or sure). It gives the probability of an ideal observer to correctly distinguish between two different states given an AIC neuron's firing rate distribution in each state. An AUC value significantly different from 0.5 indicates that the AIC neuronal response at least a partial discriminates between two conditions.

For each AIC neuron, we calculated two AUC values. First, for the AUC of explicit choice, we compared the firing rate distributions on choice trials when the monkey chose the gamble

versus the sure option. Second, for the AUC of implicit risk-attitude, we compared the firing rate distributions on choice trials when the monkey was risk-seeking versus risk-avoiding. Risk-seeking trials were defined as trials where the monkey chose the gamble, even when the expected value of the gamble option was smaller than the expected value of the sure option. We did not include trials, in which the monkey chose the gamble option when it had a higher expected value, because in that case the monkey's choice did not give any indication about his implicit risk-attitude at that moment. Conversely, risk-avoiding trials were defined as trials where the monkey chose the sure option, even so it had a lower expected value than the gamble option. Thus, trials used to compute the AUC of implicit risk-attitude were the subset of the trials, in which the monkeys did not make choices that maximized the expected value of the chosen option.

We found that trial-by-trial fluctuations in the activity of a subset of AIC neurons (35/240; 15%) significantly correlated with fluctuations of choice or risk-attitude. As shown in Fig. 5, 15

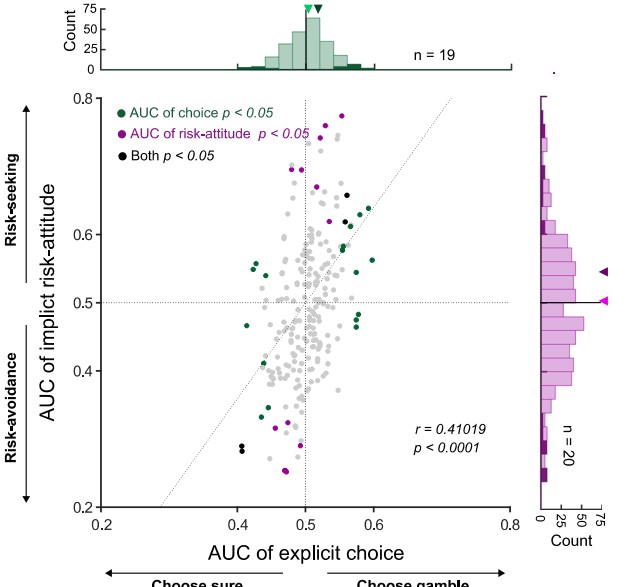

**Fig. 5 Distribution of area under the curve (AUC) of receiver operating characteristic (ROC) for explicit choice and implicit risk-attitude in individual neurons.** AUC values capturing the covariation of each neuron with differences in explicit choice (choosing gamble or sure) and implicit risk-attitude (risk-seeking or risk-avoidance). Each point represents one neuron ($n = 240$), and colors indicate the significance of the two AUC values. The gray vertical and horizontal dashed lines show the area of no significant discrimination ability (AUC of choice = 0.5 and AUC of risk-attitude = 0.5). The broken line represents the linear regression relating the AUC of explicit choice and AUC of implicit risk-attitude ($r$ and $p$ values refer to the regression slope). In the marginal distributions, significant neurons are indicted in darker shades and the arrowheads indicate the average values across the entire distribution (light green or light purple) and the subset of neurons with significant AUC (dark green or dark purple), respectively.

neurons (6%) showed a significant AUC of explicit choice (green), 16 neurons (7%) showed significant AUC of implicit risk-attitud*e* (purple), and 4 neurons (2%) showed both significant AUC of explicit choice and significant AUC of implicit risk-attitude (black). Across the AIC population ($n = 240$), the ability of neural activity to predict AUC of explicit choice and implicit risk-attitude showed a strong positive correlation (Pearson correlation; $r = 0.41$, $p < 10^{-4}$). We further examined whether neurons encoding specific-variables were particularly predictive of explicit choice or implicit risk-attitude (Supplementary Fig. 8). However, a chi-square test showed no significant dependency between the encoding of a specific decision-related variable and the likelihood that choice-predictive and/or risk-attitude-predictive signals were carried by a given AIC neuron (Table 3, $\chi2 = 7.61$, $p = 0.67$, excluding neurons with AUC that predict neither choice nor risk-attitude).

The AUC analysis showed that fluctuations in the activity of some AIC neurons are predictive of choice and/or risk-attitude. This suggests a causal role of AIC in risky decision-making. AIC neurons might encode value information, on which the monkeys' choice is based and contextual information that influences momentary risk-attitude.

## Discussion

Prospect theory provides profound insights into how humans make risky decisions in a wide range of circumstances[6,28]. The behavioral hallmarks described by the theory have also been reported in old-world and new-world monkeys, as well as in rats[22,29–31]. This suggests that the neural circuits responsible for making risky decisions may have been evolutionarily conserved across mammals. Using a token-based gambling task, we demonstrated that activity of AIC neurons in macaques exhibits critical characteristics consistent with those postulated by Prospect Theory. Decoding the activity from a subgroup of AIC neurons can predict the monkey's choices and risk attitude on a trial-by-trial basis. These results suggest that the AIC is a pivotal part of a circuit monitoring state and context information that controls risky choices by modifying activity in downstream decision processes.

Overall, monkeys in the present study were more prone to choose gamble options (Fig. 1c). This is in line with similar findings of previous monkey experiments[22,29,32,33], yet is inconsistent to most human studies[6,28,34,35]. It is unclear whether such a discrepancy between humans and monkeys was due to species-specific differences, individual variability, or task design. Macaques have been shown to be risk-aversive, like humans, in a foraging task[36] and in a risky decision-making task using animals' hydration state to index their non-monetary wealth level[3]. The observed tendency to choose gamble options was therefore likely due to task-specific factors, such as the small reward amount at stake and the large number of trials.

Insular cortex is a large heterogeneous cortex that is typically divided into posterior granular, intermediate dysgranular, and anterior agranular sectors, based on cytoarchitectural differences. Our recordings were concentrated in the most anterior part of the insula and encompassed mostly agranular and some dysgranular

**Table 3 The number of each signal during the choice period in the force choice trial, recounted based upon the AUC for explicit choice or implicit risk-attitude.**

|  | Token + value + risk | Token + value | Token + risk | Value + risk | Token | Value | Risk | End token | None |
|---|---|---|---|---|---|---|---|---|---|
| Both | 0 | 0 | 1 | 0 | 3 | 0 | 0 | 0 | 0 |
| Choice | 0 | 0 | 3 | 0 | 5 | 0 | 2 | 1 | 4 |
| Risk-attitude | 0 | 1 | 3 | 0 | 8 | 0 | 3 | 0 | 1 |
| Neither | 3 | 12 | 23 | 1 | 104 | 5 | 16 | 9 | 32 |

areas (Supplementary Fig. 6). In addition, we also recorded some neurons in the border regions of the adjacent gustatory cortex. Importantly, we found no functional segregation or gradient with respect to the functional signals that were represented across the different cortical areas, we explored. This fits with a recent primate neuroimaging study that showed strong activation of this entire region by visual cues indicating reward, as well as reward delivery[20]. Insula has long been known to be strongly connected with the neighboring gustatory cortex[37,38]. Recently, several lines of studies have demonstrated that neurons in the gustatory cortex not only engage the primary processing of gustatory inputs, but also involve multisensory integration[15,16], as well as higher cognitive functions like decision-making[13,38]. This suggests that primate insular cortex and the neighboring gustatory cortex are strongly interconnected and form an interacting distributed network during decision making.

Prospect theory assumes that people make decisions based on the potential gain or losses relative to a reference point. In our experiment, the natural reference point against which the monkey compared possible outcomes was the current token assets. Consistent with this idea, we found that the activity of a substantial number of AIC neurons (109/240; 45%) encoded start token number. The AIC has been suggested to represent the current physiological state of the subject (i.e., interoception)[15,16,39,40]. Our findings suggest that AIC also encodes more abstract state variables, such as current wealth level, which are important for economic decisions that will influence future homeostatic state. Notably, we found some AIC neurons encode the start token number in a numerical scale, with their activity increased (or decreased) specifically when the monkey owned a particular number of tokens. Such a pattern of numerical encoding has been identified in primate prefrontal and parietal cortex[41,42], medial temporal lobe[43], and recently in AIC[44]. It would be interesting for future studies to investigate whether these number-tuned neurons relate to the numerical abilities of primates.

Our results overwhelmingly support the notion that value-related signals in the brain operate in a relative framework. Only 4% of neurons in the AIC carried a value signal in an absolute framework. However, how the value of options is represented in a relative framework is an issue under debate. The core of the debate regards whether the value of gains and losses are represented in a single unitary system[45] or separately by two independent systems[46–48]. Some of the AIC neurons encoding a parametric value signal did continuously across gains and losses (14/47; 30%). However, most AIC neurons encoded gain or loss-specific value signal (33/47; 70%). Thus, while there is some evidence for both hypotheses, most AIC value-encoding neurons form two independent representations that encode gains or losses, respectively. This functional separation is further supported by the presence of a large number of neurons carrying a categorical gain/loss signal. Interestingly, the number of loss-encoding neurons (29/33; 88%) is much larger than the number of gain-encoding AIC neurons (4/33; 12%). This may explain why human imaging studies often find a link between the AIC and the anticipation of aversive outcomes[1,13]. Thus, the AIC recordings show the presence of separate neuronal populations that encode value as a relative gain or loss. This could be the neuronal underpinning of the separate utility functions used in prospect theory.

The behavior of the monkey did not only indicate that they use a relative value framework (Fig. 1e), but also that risk-attitude was modulated by the token assets (Fig. 1d). We described this effect as a state-dependent modulation of the prospect theory model (Fig. 2b–d). This asset effect might be due to deliberate strategic adjustments of risk attitude to optimize reward rate[49]. Alternatively, the asset effect might represent a contextual bias related

to wealth level. Future work will be necessary to distinguish between these different hypotheses.

Risk is often formalized and quantified as the outcome variance, and the AIC has been implicated to play a role in monitoring risk[25,26]. In line with this, we found 8% of the AIC neurons ($n = 19/240$) whose activity correlates with the outcome variance. Moreover, the trial-by-trial variability of the monkeys' choice and risk attitude was correlated with activity changes in a subgroup of AIC neurons (Fig. 5 and Supplementary Fig. 8). All of this supports the hypothesis that AIC plays an important part in the process of decision making under risk.

There are number of limitations to our results. First, in our present study we introduce only a small number of reward amount and probability levels, which made it very difficult to test for non-linearities in the utility and probability weighting function. This is true both for the behavior and for the underlying neuronal representations. Testing the hypothesis of non-linearity will require exploring a much larger parametric space for the potential outcomes and outcome probability. Second, we do not know if the neuronal signals we observed are unique to the AIC or if similar signals exist in other brain areas. Future work will require recording in brain areas that are up-stream and downstream of AIC to determine the specific role of the AIC in the neural circuit underlying risk-attitude. Third, we do not know whether the AIC plays a causal role in risky decision-making. This will require testing if and how different manipulations of the activity of AIC neurons affect animals' behavior.

In this study, we interpreted the function of AIC from the perspective of economic, risky decisions[13] and within the framework of Prospect theory[6]. Decisions are likely not only guided by the rational, abstract processes depicted by economic models, but are strongly influenced by emotional processes[50]. The AIC has been suggested to occupy a central position in regulating emotions as it receives interoceptive afferents from visceral organs through the posterior granular insula area, representing contextual information (as demonstrated by this study), and is closely connected with the amygdala and autonomic nuclei[15,51]. This study took the first step to delineate how the decision context modulates economic value representation, and thereby impacts the decision of subjects. Future work will further investigate the interacting functions of AIC in economic decisions, emotions, and autonomic regulation.

## Methods

All animal care and experimental procedures were conducted in accordance with the US public Health Service policy on the humane care and use of laboratory animals and were approved by the Johns Hopkins University Institutional Animal Care and Use Committee (IACUC).

**General**. Two male rhesus monkeys (Monkey G: 7.2 kg, 7 years; Monkey O: 9.5 kg, 8 years) were trained to perform a token-based gambling task in this study. MonkeyLogic software[52] (https://www.brown.edu/Research/monkeylogic/) was used to control task events, stimuli, and reward, as well as monitor and store behavioral events. During the experimental sessions, the monkey was seated in an electrically insulated enclosure with its head restrained, facing a video monitor. Eye positions were monitored with an infrared corneal reflection system, EyeLink 1000 (SR Research) at a sampling rate of 1000 Hz. All analyses were performed using self-written Matlab code, unless noted otherwise.

**Behavioral tasks**. The token-based gambling task was based on a previously published task design[53] and consisted of two types of trials: choice trails and forced choice trials. In choice trials, two targets (a sure option and a gamble option) were presented on the screen. Monkeys were allowed to choose one of the options by making a saccade to the corresponding target. Choice trials allowed us to measure the monkey's risk attitude in different behavioral contexts of various value-matching of gamble and sure. In forced choice trials, only one target (either a sure option or a gamble option) was presented on the screen so the monkey was forced to make a saccade to the given target. Comparing the neuronal activity in choice and forced choice trials allowed us to identify neuronal signals specifically related to decision-making. The choice and force choice trials were pseudo-randomly

interleaved in blocks so that each block consisted of all 24 choice trials and 13 force choice trials. The spatial location of the target cues indicating the options was randomized across both choice and forced choice trials.

A choice trial began with the appearance of a fixation point surrounded by the token cue. After the monkey had maintained its gaze at the central fixation point (±1° of visual angle) for a delay period (0.5–1 s), two choice targets were displayed on two randomly chosen locations among the four quadrants on the screen. The monkey indicated its choice by shifting gaze to the target. Following the saccade, the token cue moved to surround the chosen target and the unchosen target disappeared from the screen. The monkey was required to keep fixating the chosen target for 450–550 ms, after which the chosen target changed either color or shape. If the chosen target was a gamble option, it changed from a two-colored square to a single-colored square to indicate the outcome of the gamble. The color represented the amount of gained or lost tokens in the present trial. If the chosen target was a sure option, the shape changed from a square to a circle serves as a control to the change in visual display that occurs during gamble option choices. Finally, after an additional delay (500 ms) the token cue was updated. If the owned token number was equal to or more than 6 at this stage, the monkey received a standard fluid reward after an additional 450 ms waiting time. At the beginning of the next trial, the remaining tokens were displayed with filled circles. Otherwise, if the owned token number was smaller than 6, the monkey did not receive a fluid reward and the updated token cue was displayed at the beginning of the next trial. If the owned token number was smaller than 0, the intertrial-interval (ITI) for the next trial would be prolonged (300 ms per owed token).

The monkey was required to maintain the fixation spot until it disappeared for reward delivery. If the monkey broke fixation in either one of the two time periods, the trial was aborted and no reward was delivered. The following trial repeated the condition of the aborted trial contingent on the time of fixation break. A trial in which the monkey broke fixation before the choice was followed by a trial in which the same choice targets were presented, but at different locations. This ensured that the monkey sampled every reward contingency evenly but could not prepare a saccade in advance. On the other hand, a trial in which the monkey broke fixation after the choice was followed by a forced choice trial in which only the chosen target was presented. If the monkey broke fixation following a gamble choice, but before the gamble outcome was revealed, the same gamble cue was presented. If the monkey broke fixation following a sure choice or after a gamble outcome was revealed, the same sure cue was presented. This ensured that the monkey could not escape a choice once it was made and had to experience its outcome. All trials were followed by a regular 1500–2000 ms ITI. The schedule of the token-based gambling task is shown in Fig. 1a.

All options in this task were represented by sets of colored squares, with the color of the square indicating the token that could be gained or lost (token outcome) and the proportion of color indicating the probability that this event would take place (outcome probability) (Fig. 1b). The sure options were single-colored squares indicating a certain outcome (gain or loss of token). There were seven different colors used for sure options representing the number of tokens that were gained or lost ([−3, −2, −1, 0, +1, +2, +3]). The gamble options were two-colored squares indicating two possible outcomes indicated by two different colors. The portion of each color corresponded to the probability of each outcome. Six gamble options were used in this task. Three of the gambles resulted in either a gain of 3 or 0 token(s), but with different outcome probabilities (i.e., token [+3, 0] with the probability combination of [0.1, 0.9], [0.5, 0.5], or [0.75, 0.25]). Another three gambles resulted either in a loss of 0 or 3 token(s) with different outcome probability (i.e., token [0, −3] with probability combination of [0.1, 0.9], [0.5, 0.5], or [0.75, 0.25]). The choice trials were divided into a gain context and a loss context (Fig. 1b). In the gain context, the three gamble options that resulted in either a gain of +3 or 0 token with different outcome probabilities were paired with four sure options that spanned the range of gaining outcomes (i.e., [0, +1, +2, +3]). These resulted in 12 possible combinations of sure and gamble options. In the loss context, the other three gamble options resulted either in a loss of 3 or 0 with probability combination were paired with four sure options that spanned the range of losing outcomes (i.e., [0, −1, −2, −3]). Thus, there were another 12 possible combinations of sure and gamble options. This resulted in a total of 24 different combinations of reward option combinations (half in the gain context and the other half in the loss context) that are offered in choice trials. In the force choice trials, all 13 different reward options (7 sure and 6 gamble options) which were used in the choice trials are presented in isolation.

**Saccade detection**. Eye movements were detected offline using a computer algorithm (saccade detection function) that searched first for significantly elevated velocity (30°/s). Saccade initiations were defined as the beginning of the monotonic change in eye position lasting 15 ms before the high-velocity gaze shift. A valid saccade for choice was further admitted to the behavioral analysis if it started from the central fixation window (1° × 1° of visual angle) and ended in the peripheral target window (2.5° × 2.5° of visual angle).

**Description of monkeys' behavior**. Fixation behavior: We examined whether and how monkeys' motivations to initiate a new trial were influenced by the outcome of the previous trial and the start token number of the current trial. We used two

behavioral signals as indications of the monkey's motivational state: (1) fixation latency (i.e., the time from fixation point onset until fixation by the monkey) and (2) fixation break ratio (i.e., the frequency with which the monkey failed to fixate on the fixation point long enough to initiate target onset). We used linear regression models to test if there was a significant relationship between each of the two variables describing motivational state and the variables describing history and current state.

Response time: We examined whether and how response times were influenced by different decision-related variables. For each trial, response time was defined as the time period between target onset and saccade initiation estimated by the saccade detection function. The response time dataset in each condition (e.g., trials from context of gain or loss, trials with different start token numbers, trials with different expected values of chosen option (chosen EV), or trials with different absolute values of difference of expected values among the gamble and sure option (|ΔEVgs|)) was fitted with an ex-Gaussian distribution algorithm[54] (https://doi.org/10.6084/m9.figshare.971318.v2). It returned three best-fitting parameter values of the ex-Gaussian distribution: the mean $\mu$, the variance $\sigma$, and the skewness $\tau$ of the distribution. We used a one-sided permutation test to determine if the mean RTs of trials from the gain and loss context are significantly different. We used linear regression models to test whether there was a significant relationship between mean RTs and start token number, chosen EV, or |ΔEVgs|.

**Prospect theory model**. Prospect theory is derived from classical expected value theory in economics[55] and assumes that the expected utility of a gamble depends on the utility of the reward amount that can be earned, weighted by the subjective estimation of the probability of the particular outcome. Both the utility function and the probability function can be non-linear and thus might influence risk preference.

We modeled the probability that monkeys chose the gamble option by a softmax choice function whose argument was the difference between the expected utility of each option.

Utility was parameterized as:

$$u(x) = \begin{cases} x^\alpha, & \text{for } x > 0 \\ -\lambda(-x)^\alpha, & \text{for } x < 0 \end{cases}, \tag{1}$$

where $\alpha$ is a free parameter determining the curvature of the utility function, $u(x)$, and $x$ is the reward outcome (in units of gaining or losing token numbers). $\lambda$ indicates the modulation of utility function in loss context.

Subjective probability of each option is computed by:

$$w(p) = \frac{p^\gamma}{\left(p^\gamma + (1-p)^\gamma\right)^{1/r}}, \tag{2}$$

where $\gamma$ is a free parameter determining the curvature of the probability weighting function, $w(p)$, and $p$ is the objective probability of receiving corresponding outcome.

The expected utility (EU) of each option was computed by combining the output of $u(x)$, and $w(p)$ that map objective gains and losses relative to the reference point and objective probability onto subjective quantities, respectively:

$$\text{EU}_{\text{gamble}} = u(x_{\text{win}}) \times w(p_{\text{win}}) + u(x_{\text{loss}}) \times w(1 - p_{\text{win}}), \tag{3}$$

$$\text{EU}_{\text{sure}} = u(x) \times w(1). \tag{4}$$

The expected utility difference between the two options was then transformed into choice probabilities via a softmax function with terms of slope $s$ and bias $s$:

$$P(\text{Gamble}) = \frac{1}{1 + e^{-s(\Delta\text{EU}-\text{bias})}}, \tag{5}$$

where $\Delta\text{EU} = \text{EU}_{\text{gamble}} - \text{EU}_{\text{sure}}$, $s$ determines the sensitivity of choices to the $\Delta\text{EU}$, and bias is the directional bias of choosing gamble.

For an alternative expected value (EV) model, the value of option is calculated as:

$$\text{EV}_{\text{gamble}} = x_{\text{win}} \times p_{\text{win}} + x_{\text{loss}} \times (1 - p_{\text{win}}), \tag{6}$$

$$\text{EV}_{\text{sure}} = (x) \times (1). \tag{7}$$

The expected value difference between the two options was then transformed into choice probabilities via a softmax function with terms of slope $s$ and bias $s$ as what we did in the PT model:

$$P(\text{Gamble}) = \frac{1}{1 + e^{-s(\Delta\text{EV}-\text{bias})}},$$

where $\Delta\text{EV} = \text{EV}_{\text{gamble}} - \text{EV}_{\text{sure}}$, $s$ determines the sensitivity of choices to the $\Delta\text{EV}$, and bias is the directional bias of choosing gamble.

We optimized model parameters, $\alpha$, $\lambda$, $\gamma$ in the PT model, and $s$ and bias in both PT and EV models by minimize the negative log likelihoods of the data given different parameters setting using Matlab's fmincon function, initialized at multiple

starting points of the parameter space as follows:

$$0 < \alpha, \lambda, \gamma < 5,$$

$$-10 < \text{bias} < 10$$

$$0 < s < 20$$

−2 negative log-likelihoods ($-2 * LL_{max}$, which measures the accuracy of the fit) were used to compute classical model selection criteria. We also computed the Bayesian information criterion (BIC):

$$\text{BIC} = \log(n) * \mathrm{d}f - 2 * \text{LL}_{max},$$

where $n$ is the number of training trial and $\mathrm{d}f$ is the number of free parameters in the model. The likelihood in BIC is penalized by adding more parameters into the model. Thus, we use BIC to represent the trade-off between model accuracy and model complexity and use it to guide model selection. We then compared $-2 * LL_{max}$ and BIC calculated from a 5-fold cross-validation with separate training and testing for the PT model and EV model in one-sided paired $t$-tests.

As in classical expected value theory in economics[56], a convex utility function ($\alpha > 1$) implies risk seeking, because in this scenario, the subject values large reward amounts disproportionately more than small reward amounts. Gain from winning the gamble thus has a stronger influence on choice than loss from losing the gamble. In the same way, a concave utility function ($\alpha < 1$) implies risk seeking, because large reward amounts are valued disproportionally less than small ones.

The $\lambda$ can further influence subject's risk-attitude in the context of gain or loss because it modulates the curvature of utility function in gain to that in the loss context. With a $\lambda > 1$, the utility function in the loss context will be sharper than that in the gain context, indicating the subject is more sensitive to the value change in the loss context. While with a $\lambda < 1$, the utility function in the loss context will be flatter than that in the gain context, indicating the subject is less sensitive to the value change in the loss context.

Independently, a non-linear weighting of probabilities can also influence risk attitude. For example, an S-shaped probability weighting function ($\gamma < 1$) implies that the subject overweighs small probabilities and underweights the large probabilities. This would lead to higher willingness to accept a risky gamble, because small probabilities to win large amounts would be overweighed relative to high probabilities to win moderate amounts.

The bias term in the softmax function can also influence a subject's risky choices independent to the expected utility of options. A negative bias will result with risk-seeking behavior because the subject tends to choose gamble while the EUs of gamble and sure are identical. In the other hand, a positive bias will result with risk-aversive behavior because the subject tends to choose sure while the EUs of gamble and sure are identical.

**Cortical localization and estimation of recording locations**. We used T1 and T2 magnetic resonance images (MRIs) obtained for the monkey (3.0 T) to determine the location of the anterior insula. In primates, the insular cortex constitutes a separate cortical lobe[57], located on the lateral aspect of the forebrain, in the depth of the Sylvian or lateral fissure (LF) (Fig. 3a, Supplementary Fig. 6). It is adjoined anteriorly by the orbital prefrontal cortex, and it is covered dorsally by the frontoparietal operculum and ventrally by the temporal operculum. The excision of the two opercula and part of the orbital prefrontal cortex reveals the insula proper, delimited by the anterior, superior, and inferior peri-insular (or limiting or circular) sulci. We used the known stereoscopic recording chamber location and recording depth of the electrode to estimate the location of each recorded neurons. The estimated recording locations were superimposed on the MRI scans of each monkey. Cortical areas were estimated using the second updated version of the macaque monkey brain atlas by Saleem and Logothetis[58] with a web-based brain atlases[59].

**Surgical procedures**. Each animal was surgically implanted with a titanium head post and a hexagonal titanium recording chamber (29 mm in diameter) 20.5 mm (Monkey G) and 16 mm (Monkey O) lateral to the midline, and 30 mm (Monkey G) and 34.5 mm (Monkey O) anterior of the interaural line. A craniotomy was then performed in the chambers on each animal, allowing access to the AIC. The location of AIC was determined with T1 and T2 magnetic resonance images (MRIs, 3.0 T) obtained for each monkey. All sterile surgeries were performed under anesthesia. Post-surgical pain was controlled with an opiate analgesic (buprenex; 0.01 mg/kg IM), administered twice daily for 5 days postoperatively.

**Neurophysiological recording procedures**. Single neuron activities were recorded extracellularly with single tungsten microelectrodes (impedance of 2–4 MΩs, Frederick Haer, Bowdoinham, ME). Electrodes were inserted through a guide tube positioned just above the surface of the dura mater and were lowered into the cortex under control of a self-built Microdrive system. The electrodes penetrated the cortex perpendicular to the surface of the cortex. The depths of the neurons were estimated by their recording locations relative to the surface of the cortex. Electrophysiological data were collected using the TDT system (Tucker & Davis). Action potentials were amplified, filtered, and discriminated conventionally with a

time–amplitude window discriminator. Spikes were isolated online if the amplitude of the action potential was sufficiently above a background threshold to reliably trigger a time–amplitude window discriminator and the waveform of the action potential was invariant and sustained throughout the experimental recording. Spikes were then identified using principal component analysis (PCA) and the time stamps were collected at a sampling rate of 1000 Hz.

**Spike density function**. To represent neural activity as a continuous function, we calculated spike density functions by convolving the peri-stimulus time histogram with a growth-decay exponential function that resembled a postsynaptic potential[60]. Each spike therefore exerts influence only forward in time. The equation describes rate as a function of time $R(t)$, $R(t) = (1 − \exp(−t/\tau g)) * \exp(−t/\tau d)$, where $\tau g$ is the time constant for the growth phase of the potential and $\tau d$ is the time constant for the decay phase. Based on physiological data from excitatory synapses, we used 1 ms for the value of $\tau g$ and 20 ms for the value of $\tau d$[61].

**Decision variables used for regression analysis of neuronal coding**. To quantitatively characterize the variables that each AIC neuron encodes during the choice period, we performed a series of multiple linear regressions. We tested if the neuronal activity was related to three types of decision-related variables: token-related variables, value-related-variables, and risk-related variables.

Token-related variables represented the start token number. We tested three types of variables. The first type, the Parametric token signal, encoded the start token number in a linear, continuous manner (monotonically rising or falling from 0 to 5). The second type, the Categorical token signal, encoded the start token number in a binary, discontinuous manner (with a value of 1 for trials with start token number 0 to 3 and a value of 2 for trials with start token number 3–5). The third type, the Numerical token signal, encoded the start token number in a Gaussian manner with the peak of the activity at one of the token numbers from 0 to 5, and the activity symmetrically falling for token numbers that were smaller or larger than the peak, generating six distinct models to be tested.

Value-related variables represented the value of options. We tested five types of variables. The first type, the Gain/Loss signal, encoded the context of gain or loss in a binary manner. Trials in the gain context were indicated with a 1, and trials in the loss context with a −1. The only exception were forced choice trials with a sure option with EV = 0, which were indicated with a 0. The second type, the General value signal, encoded the expected value of options in a linear, continuous manner across both the gain and loss context (with a range from −3 to 3). The remaining types also encoded the expected value of options, but contingent on the gain/loss context. The third type, the Gain value signal, encoded the expected value of options in a linear manner, but only in the gain context (options with EV larger than 0 were encoded as the original number, otherwise were encoded as 0), while the fourth type, the Loss value signal, encoded the expected value of options in a linear manner only in the loss context (options with EV smaller than 0 were encoded as the original number, otherwise were encoded as 0). The fifth type, the Behavioral salience signal, encoded the expected value of options in a linear but asymmetric direction for the gain and loss context. Thus, this signal encoded the absolute distance of the value from zero, independent of whether it represented a gain or a loss (e.g., both an option with EV = 1.5 and an option with EV = −1.5 would be encoded as 1.5).

Risk-related variables represented the risk associated with an option. Risk was defined as the variance of possible outcomes of an option (calculated by $\sqrt{p(1−p)}$, in which $p$ was the winning probability of the option). We considered two different types. The first type, the Parametric risk signal, encoded the variance of outcome in a linear manner proportional to the variance. The second type, the Categorical risk signal, encoded whether the outcome of option was uncertain or not in a binary manner (with a value of 1 for all gamble options and a value of 0 for all sure options).

For all value-related and risk-related variables, outcomes (and their values) were always defined as a relative change of tokens, independent of the start token number. We also considered an End token signal (i.e., the sum of all possible end token numbers, weight by their probability). This signal took into account not only the possible change in token number, but also the start token number. Thus, it represented the outcome of a choice in an absolute framework that reflected how close the monkey was to earning fluid reward.

**Mix-selective neuronal coding with regression analysis**. To quantitatively characterize the variables that each AIC neuron encodes, we examined the activity of each neuron using a series of multiple linear regression models, with the mean Firing rate (FR) within the choice period for each trial as the dependent variable, and predictors derived from the decision-related variables as the independent variables:

$$\text{FR} = \text{constant} + \alpha1 * \text{predictor}1 + ...... + \alpha i * \text{predictor } i$$

We tested all potential combinations of three types of basic variables (token, value, and risk) and a baseline term. For each of the basic variable types, we tested each model variant with a specific decision variable belonging to that type (eight forms of token-encoding, five forms of value-encoding, and two forms of risk-encoding). Within such a family of models, a particular basic variable type (e.g.,

value) could only be represented by one specific decision variable (e.g., a model that contained a value-related variable could include Gain value or Gain/Loss), but not by combinations of specific variables (e.g., Gain/loss + Gain value was excluded). This constraint was chosen because testing all possible combinations of our 15 decision variables would have resulted in $2^{15} = 32,768$ models that needed to be tested for each model. Our data would not have allowed a robust differentiation of such a large number of models.

The restricted combination of decision variables resulted in a total of 80 three-variable encoding models (8 * 5 * 2), 66 two-variable models (8 * 5 + 2 * 5 + 2 * 8), and 15 single variable models (8 + 5 + 2). In addition, we tested a model that included only End token number as a variable and a model containing only a baseline term. Thus, in total, we tested 163 models for each neuron. We determined the best-fitting model for each neuron using the Akaike information criterion and classified it as belonging to different functional categories according to the variables that were included in the best-fitting model. We computed the Akaike information criterion (AIC):

$$AIC = 2 * df - 2 * LL_{max},$$

where $df$ is the number of free parameters in the model and $LL_{max}$ is the log-likelihood.

**Sensitivity to value changes in gain and loss value cells**. We examined whether neurons carrying Loss value signals and Gain Value signals showed different sensitivity to changes in value. We estimated the absolute value of the standardized regression coefficients (|SRCs|) of firing rate of Loss-Value Neurons in the loss context and |SRCs| of firing rate of Gain-Value Neuron in the gain context, respectively. We included all neurons, whose best-fitting model included a Loss or Gain Value Signal. We performed a one-sided permutation test with 10,000 iterations to test if the normalized |SRCs| of Loss Value ($n = 39$) and Gain Value neurons ($n = 12$) showed a significant difference (Fig. 4a).

We also examined whether the sensitivity to value change in neurons carrying Loss value signals and Gain Value signals was influenced by the wealth level (i.e., the number of tokens owned at the beginning of the trial). We compared |SRCs| of Loss and Gain Value neurons in trials with low [0, 1, 2] or high [3, 4, 5] token level. We performed a one-sided permutation test with 10,000 iterations to test if the normalized |SRCs| of Loss Value ($n = 39$) and Gain Value neurons ($n = 12$) showed a significant difference for low and high token levels (Fig. 4c).

**Receiver operating criterion (ROC) analysis**. To determine whether neural activity of the AIC neurons was correlated with the monkey's choice behavior or risk-attitude, we computed a ROC for each cell and computed the AUC as a measure of the cell's discrimination ability. An AUC value near 1 indicates an almost perfect association between neuronal responses and state (e.g., the neuron generates two non-overlapping firing rate distributions: a higher firing rate distribution when choosing the gamble option [state 1] and a lower firing rate distribution when choosing the sure option [state 2]), so that the ideal observer can predict the behavioral state (e.g., the choice) near perfectly based on the neuronal activity. On the other hand, an AUC value near 0.5 indicates no clear correlation between neuronal responses and state (e.g., two overlapping FR distributions when choosing the gamble and sure option) so that the ideal observer exhibits chance performance. Thus, an AUC value significantly different from 0.5 indicates at least a partial discrimination between two conditions. We computed the AUC value of explicit choice by comparing the distributions of firing rates associated with each of the two choices (i.e., choice of gamble or choice of sure). We computed the AUC value of implicit risk-attitude by comparing the two distributions of firing rates associated with risk-seeking and risk-avoidance behavior. Risk-seeking trials were defined as trials where the monkey chose the gamble, even so the expected value of the gamble option was smaller than the expected value of the sure option. We did not include trials, in which the monkey chose the gamble option and it had a higher expected value, because in that case the monkey's choice did not give any indication about his risk-attitude at that moment. Conversely, risk-avoiding trials were defined as trials where the monkey chose the sure option, even so the expected value of the sure option was smaller than the expected value of the gamble option. Thus, trials used to compute the risk-seeking probability were a subset of the trials used to compute choice probability, in which the monkeys did not make choices that maximized the expected value of the chosen option. The expected value difference between gamble and sure option (ΔEV) itself was correlated with the probability of choosing the gamble option (P(Gamble)). Therefore, neurons encoding option value could have high AUC values, even if they did not predict choice itself. To avoid such unreliable AUC estimations, we only include choice trials with expected value difference between gamble and sure option (ΔEV) that resulted in variable choices. We defined sufficient variability as a probability of choosing the gamble option (P(Gamble)) between 0.25 and 0.75 (trials with ΔEV > −1.5 and ΔEV < 0.25).

**Chi-square test**. To test whether neurons encoding specific behaviorally relevant variables were more likely to carry significant choice or risk-attitude probability signals, we used a chi-square test, which is used to determine whether there is a statistically significant difference between the expected frequencies and the observed frequencies in one or more categories.

**Reporting summary**. Further information on research design is available in the Nature Research Reporting Summary linked to this article.

## Data availability

Source Data are provided with this paper for all data presented in graphs within the figures. The raw data of behavior and neurophysiology are used for other papers under preparation. Access can be obtained upon reasonable request from the Lead Contact, Veit Stuphorn (veit@jhu.edu). Source data are provided with this paper.

## Code availability

The code used for data analysis is available upon reasonable request from the Lead Contact, Veit Stuphorn (veit@jhu.edu).

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

## Acknowledgements

The authors would like to thank Bill Nash and Bill Quinlan for technical help, and Xiaomo Chen, Jacob Elsey, Christof Fetsch, Jim Knierim, Kishore Kuchibhotla, Ernst Niebur and Wen-Kai You for comments on the manuscript. Special thanks to Daeyeol Lee for feedback on the paper and long discussions about the project. This work was supported by the National Institutes of Health through grant R01DA049147 to V.S.

## Author contributions

V.S. designed the experiment, X.L. trained the monkeys, and recorded behavioral and neural data, Y.Y. analyzed the data, V.S. and Y.Y. wrote the paper.

## Competing interests

The authors declare no competing interests.
