## [Peer Review File · Nature Communications]

Primate anterior insular cortex represents economic decision variables postulated by Prospect theoryREVIEWER COMMENTS

Reviewer #1 (Remarks to the Author):

The experiment is one where the agent has to sequentially choose the right options (gambles) in order to cumulate points to the right ("target") level before a reward is given. This is a stochastic dynamic programming problem studied in the neuroscience literature, among others, in a 2010 article in *Journal of Neuroscience* (Symmonds *et al.*). The 'target' generates an endogenous reference point that moves with previously acquired points (wealth). As the JN article shows, when taking each trial independently, revealed risk aversion changes with acquired points, even if the agent's (global) risk aversion is constant (Fig. 3). This is merely a consequence of the observed ignoring the overall goal of the agent, which is to reach the right target, and therefore, obtain the reward. The article, a human neuro-imaging study, reports significant activation in anterior insula to gamble variance.

Remarkably, but in line with human evidence, the monkeys in the experiment behave optimally. E.g., monkeys decrease risk aversion locally when far away from the target, but increase it when the target is reached or about to be reached.

The specific setup was taken from Seo and Lee, *Journal of Neuroscience*, 2009. There is, however, an important difference. Unlike here and in Symmonds *et al.* (2010), uncertainty was resolved through play in a biased game of matching pennies. The monkey played against a computer that changed its strategy to exploited non-Nash behavior. Importantly, unlike here and in Symmonds *et al.* (2010), pure strategies (always playing the SAME given the points accumulated and gambles presented) are NEVER optimal. As an aside, Seo and Lee (2009) also report that their subjects behaved nearly optimally. This is quite remarkable since Seo and Lee assume risk neutrality. Presumably, learning (to play the right strategy) dominates any effect of risk aversion.

Contrary to the claims of the authors, the observed behavior has NOTHING to do with prospect theory. Prospect theory is relevant to situations where a reference point appears to be used when expected utility theory prescribes that no reference point should be used. In Kahneman and Tversky's famous experiment (*Econometrica*, 1979), IDENTICAL gambles ("prospects") are proposed, but framed differently. The frames make subjects think in terms of an imaginary reference point, and losses and gains. Behavior changes when the frame, and hence, the reference point, is manipulated. In contrast, in the task presented here the reference point is not imaginary. It is REAL. The "frame" is a target that moves depending on how many points have already been accumulated.

The only semblance of prospect theory in the monkeys' behavior is overweighing of small probabilities. This has been observed for monkeys before: Stauffer *et al.*, *Journal of Neuroscience* 2015 (Reference 26 in the paper). This in itself, however, does not violate expected utility axioms. Agents are allowed to employ subjective beliefs, as long as they do not violate principles of rational choice such as the sure thing principle. Nothing in the reported results suggests that the monkeys made irrational choices.

Incidentally, in the analysis of the experiment on which the present one builds, Seo and Lee (2009), there is NEVER any mention of prospect theory. This is as it should be. Seo and Lee do analyze the difference in behavioral and neural effects after gains and losses, but this is with a focus on LEARNING (to play the right strategy). By then, it was well known that animal learning in gain and loss conditions differ. Prospect Theory does not cover learning. Prospect Theory is about choices in stable conditions, when the probabilities of the outcomes are (supposed to be) known. Among others, it is well known that the usual inverted-S shaped probability weighting from Prospect Theory (confirmed here and in Stauffer *et al.*, 2015) flips to an S-shaped function under learning (see the discussion in Hertwig and Erev, *TICS* 2009).

I found the imaging results most interesting, and very well presented. I would urge the authors however to refrain from presenting their case as a study of prospect theory, and follow prior literature in correctly interpreting their task as one of stochastic dynamic programming rather than prospect theory.

I will not be able to formulate a definite recommendation about this paper until the authors do so. An interpretation based on prospect theory is misleading. It is analogous to interpreting outer space images with false colors where the false colors are not even inspired by position on the spectrum.

Reviewer #2 (Remarks to the Author):

Abstract:

- the token task is not all that novel and was used by Lee before (2009 JN). This is mentioned later, but in the Abstract the word 'new' should be removed.
- what the authors label as 'asymmetric value function' is correct but is more commonly and specifically referred to as 'loss aversion' (steeper loss than gain function, mentioned in Results section with fig 1). The authors may consider to rather use the term of loss aversion that is more specific than 'asymmetric value function'.

Fig 1b:

- quotation of this fig is reversed in the Results text: gain is actually shown at left and loss at right. (I would prefer to start the description of the choice options with gains for convenient relation to previously mostly studied gains, so the sequence would be fine as in the text but should be reversed in fig 1b.)
- it is unclear what exactly were the two options the animal was presented with: the text says 'choose between a certain token increase versus an uncertain option that could result in an even larger increase or no increase', and fig 1b does not contain that information either. And by 'increase versus even larger increase or no increase', do you mean a large safe reward vs. a gamble with one larger and one smaller reward than the safe option? Please clarify. Correspondingly for losses. And please add this information into fig 1b.
- please label probabilities in the legend as $p = [0,1]$, not in %, as per general convention.

Fig 1f: what the authors call subjective value (SV) is usually called Expected Utility in standard economic notation [$EU = \sum (u(x) \cdot \pi(p))$]. Please stay with accepted terms for clarity throughout the paper.

Risk attitude (fig 1). These are very nice tests and fabulous results. It is a bit surprising that the animals were generally more risk seeking with gains than with losses, but the important thing is that they show loss aversion (steeper loss than gain function; at least monkey G) and that risk attitude changed with increasing gains towards aversion (concave utility). It is also reassuring to see that their gain utility function became less convex and more concave with higher start amounts (wealth, fig 1g top).

Line 143: the diction 'while subjects exhibit diminishing marginal sensitivity' sounds a bit odd: concave utility indicating risk aversion IS DUE TO diminishing marginal utility (not 'sensitivity', unless you want to diverge into Weber's law). One could just say 'due to diminishing marginal utility'.

Probability weighting functions: please cite the specific weighting function you are using (Prelec, Gonzalez-Wu or Kahneman-Tversky) to link the reader to proven methods. You may also consider citing one or both behavioral probability weighting studies by Stauffer et al. 2015 and/or Ferrari-Toniolo et al. 2019 (both J Neurosci) that present more details and support your behavioral analysis and conclusions.

Paragraph starting on line 180 and fig S3: a comparison between a linear EV model and a nonlinear utility model should, in such a sophisticated study as presently, be done via choices rather than the less sensitive reaction times: would the choices be better explained by EV or utility (with or without probability weighting)? Using reaction times for such a comparison is interesting but a step down from standard-of-the-field fits to choices.

Line 181: utility IS subjective by definition: 'subjective utility' might sound odd as it implies that

there is an objective utility, which does not exist by definition.

Neuronal data section starting on line 185: the variety of neuronal modulation types is bewildering (eg. Lines 195-203, which are difficult to understand for me), and the description is a bit scattered across several paragraphs and intermingled with anatomical location. It is also unclear how many neurons were tested in no-choice trials and how many in choice trials. I think it would help to make an initial overview with clear definitions, descriptions and numbers of each modulation type, also referring to Table 1 (which is appropriately labelled as 'model comparison' but does not inform about the character of the many modulation types) and Figure 2. Maybe make a succinct paragraph for each neuron type, with definitions and numbers of neurons from Table 1 and reference to the relevant parts of fig. 2, with longer paragraphs for the more numerous or more important neuron types. Then go into details of the gain-loss sensitive neurons (fig 3).

Figure 2: it would be reassuring to have a raster plot for at least one modulation type. And replace 'linear value' in the title of part c by 'monotonic value' or something similar, to avoid the confusion stated in my next paragraph.

Line 211: in a paper in which utility functions are fitted to choices, the term of 'AIC neurons reflected information about the expected value of the options' can have two meanings: (1) if the neurons are suggested to code expected value ($x \cdot p$) as opposed to expected utility ($u(x) \cdot p$, or $u(x) \cdot \pi(p)$), one would need to show a significantly better regression fit with expected value than expected utility. I don't see these data in the paper, nor an analysis that would make this distinction. (2) if the neurons are simply increasing their activity monotonically, a linear regression could be significant, as could be a rank correlation (eg. Spearman), but the result should not be interpreted as linear coding as opposed to non-linear coding if the comparison has not been made. It seems to me that the latter was the case, as line 213 correctly says 'monotonically rising'. Lack of analysis for expected utility coding vs. expected value coding would be fine given the complexity of the analyses already, but the language is important.

Paragraph starting on line 308: what does it mean when a neuron has a 'significant choice probability'? In the paragraph above, ROC AUC discriminability is used as definition of choice probability. I appreciate this comes from ROC theory, but it is worth explaining the intuition behind this in a paper dealing with probability of empirically measured behavioral choice ($p(\text{choice})$). – If this is properly explained, the result seems that all types of neuronal modulation can predict either the choice or the risk attitude, which is a very nice result worth being more explicitly presented.

Line 289: unclear what the term 'predict the internal state' contributes / means here?

Anyway, congratulations for a nicely pioneering study with solid conceptual background rarely found in neurophysiology of this kind.

Wolfram Schultz

Reviewer #3 (Remarks to the Author):

This manuscript investigated choice behavior in macaque monkeys when they choose under conditions where the potential outcomes are either gains and losses, and recorded responses from anterior insula cortex (AIC) neurons. The main results found that, similar to humans, monkeys weigh risk differently when faced with potential gains or losses, and that AIC neurons encode reference points for this gain/loss effect, as well as reference-dependent values, suggesting a role for AIC in decision-making under risk. Overall, I think this is a well-designed study with rigorous analyses. The results are important to the field because there is very little known about the neurophysiology of AIC. For instance, the finding that there is a relative prevalence of loss encoding among these neurons is important, as it is relatively uncommon to find valence selectivity at the level of cortical regions, and likely relates to findings on fMRI of insula activation in response to negative events/information.

I only have a few comments regarding the results and interpretation. First, the authors note that their recording locations were near gustatory cortex, which is in the anterior insula. Although this study didn't set out to test gustatory responses, was there any evidence for this? For example, were there neurons that responded when the animal received the primary reward?

Second, a difference between this task design and typical designs to test prospect theory in humans is that in humans, the same gamble can be framed as a gain or loss with language whereas here the gambles are actually for different outcomes. I think this distinction should be discussed in a little more depth, as well as any relevant studies from the human literature in which similar loss-only and gain-only contexts were tested and compared.

Response to Reviewers

We thank all three reviewers for their thoughtful comments on our manuscript. Here we provide a point-by-point response to each reviewer's comments along with a description of how the paper was revised accordingly. Our response is indicated in blue.

Reviewer #1 (Remarks to the Author)

We thank Reviewer #1 (R1) for the thoughtful and insightful comments on our manuscript. While clearly not convinced that Prospect Theory is the correct conceptual framework to understand our experiment, R1 still acknowledged that our neuronal data are of interest and that our findings are clearly presented. We thank the reviewer for this assessment.

We agree that R1 did bring up an important question with regards to the understanding of the monkey's behavior. However, we hope to convince the reviewer that (1) this concern does not invalidate the use of Prospect Theory as a model for analyzing the monkey's behavior and that (2) our main findings in the anterior insular cortex (AIC) are still can be best understood as supporting the basic conceptual framework used in Prospect Theory.

Specific comment by Reviewer #1:

The experiment is one where the agent has to sequentially chose the right options (gambles) in order to cumulate points to the right ("target") level before a reward is given. This is a stochastic dynamic programming problem studied in the neuroscience literature, among others, in a 2010 article in Journal of Neuroscience (Symmonds ea). The 'target' generates an endogenous reference point that moves with previously acquired points (wealth). As the JN article shows, when taking each trial independently, revealed risk aversion changes with acquired points, even if the agent's (global) risk aversion is constant (Fig. 3). This is merely a consequence of the observed ignoring the overall goal of the agent, which is to reach the right target, and therefore, obtain the reward. The article, a human neuro-imaging study, reports significant activation in anterior insula to gamble variance.

Remarkably, but in line with human evidence, the monkeys in the experiment behave optimally. E.g., monkeys decrease risk aversion locally when far away from the target but increase it when the target is reached or about to be reached.

R1 feels that prospect theory is not the correct framework for understanding the monkey's behavior in our experiment. Instead, R1 suggest that the task requirements and the behavior of the monkeys are best interpreted as a 'stochastic dynamic programming task', in which monkeys dynamically change risk-attitude to maximize the chance to receive reward. In this interpretation, the token cues serves as an indicator of how far the monkey is away from receiving the next reward. In suggesting this

interpretation, R1 concentrates on our finding that monkeys show different risk attitude for different start token number. This is interpreted as evidence that ‘monkeys decrease risk aversion locally when far away from the target but increase it when the target is reached or about to be reached’.

However, in doing so, R1 overlooks the other behavioral findings in our experiment, which support the validity of prospect theory in general and as a conceptual framework for understanding the monkeys’ choice behaviors in our task specifically.

In its strong form, the ‘stochastic dynamic programming’ hypothesis of R1 implies that the monkey uses an ‘absolute’ reference system for representing value, whereby only the final outcome of a choice matters (i.e., the expected end token number), because it indicates how close the monkey is to receiving a fluid reward. In such a framework, it should not matter if a given final token number is reached by gaining or losing tokens. In contrast, Prospect Theory uses a ‘relative’ reference system, where outcomes are represented as changes in value relative to a reference point (i.e., gains or losses of tokens relative to the current start token number). Both value representation systems are valid, but the question is, which value framework is used.

On this point, our results are clear. Whether offers entail token gains or losses has a significant effect on monkeys’ preferences. The support for this finding comes from three different findings: 1) a new behavioral analysis that compares choice trials with outcomes that have identical absolute value (i.e., identical end token numbers), but different relative value (because they represents gains or losses) shows that monkeys use a relative value framework, 2) a number of reward maximization models inspired by the suggestion of R1 underperform the prospect theory model we showed in the manuscript, and 3) the majority of AIC neurons encoded value in a ‘relative’ rather than an ‘absolute’ reference system. We describe these findings in detail in the following sections.

First, in a new behavioral analysis we tested directly, if monkeys used an absolute or a relative value reference frame to guide their choices. We compare monkey’s risk attitude toward choice options that lead to identical outcomes (i.e., identical end token number), but resulted from either gaining or losing tokens. For example, a gain trial at start token number 0 with the gamble option [0, $p=0.5$; +3, $p=0.5$] versus the sure options [0, +1, +2, +3] has the same probability distribution of expected end token numbers as a loss trial at start token number 3 with the gamble option [-3, $p=0.5$; 0, $p=0.5$] versus the sure options [-3, -2, -1, 0] (**Suppl. Figure 4a,b**; see figure below). Similar pairings can be found for two other combinations of start token numbers and gamble options leading to gains and losses. We can determine the subjective value (SV) of the gain/loss gamble options in units of end token number by estimating the choice function for the gain and loss trials and determining the certainty equivalent of each gamble option (**Suppl. Figure 4d**). Because the outcomes are identical within an absolute framework (i.e., the expected end token numbers are the same), there should not be any substantial difference in subjective value, if the monkey uses an absolute value framework. However, we found that the SV of gamble options leading to gains were significantly higher than the SV of gamble options leading to losses (**Figures 1e** and **Suppl. Figure 4c-d**; permutation test;

monkey G & O: $p < 10^{-4}$ for gamble options that resulted with the same end token number from either gaining or losing tokens). This result strongly suggests that monkeys evaluate options using a relative value framework.

These new results are described in the revised manuscript (lines 131-148) and shown in Fig. 1e and Supplementary Fig. 4 (shown below).

Supplementary Figure 4. Monkeys' risk preferences changed across gain/loss contexts with different start token numbers cannot be attributed to their evaluation of each available option based on the expected final token number. Prospect theory implies the use of a 'relative value' framework, where the value of an outcome depends on the change in assets relative to a reference point. Alternatively, the monkeys could use an 'absolute value' framework, where the value of an outcome depends on the final asset number. To test, which of these value frameworks is used by the monkeys, we compared trials with the same outcome in terms of final token number, but which resulted from either gaining or losing tokens, because the start token number was different. **(a)** For example, consider a trial with a start token number of 0, in which a gamble option with an equal probability of gaining 3 or 0 tokens is offered versus a sure option of gaining 2 tokens. The expected end token outcomes of this trial (owning 3 or 0 tokens, each with $p=0.5$ versus owning 2 tokens with $p=1$) are identical to a trial with a start token number of 3, in which a gamble option with an equal probability of losing 3 or 0 tokens is offered versus a sure option of losing 1 token. **(b)** The same two gambles can be systematically matched with other pairs of sure options that reached the same final token number by either gaining or losing tokens. The dotted boxes shows the probability distribution of expected end token numbers (i.e., the possible final token numbers and their probability) for each of the gambles and a set of corresponding sure options. The probability distribution of expected end token number for gain trials at a start token number of 0 (green dotted box) is identical to the one for loss trials at a start token number of 3 (red dotted box). Thus, the subjective value of the two gamble options should be identical in an absolute value framework, but different in a relative value framework. **(c)** Choice functions indicate the probability of choosing the gamble option as a function of expected end token number for the paired sure options. The expected end token number is a function of the indicated change in number of token and the start token number (e.g., expected end token number is 2 when choosing sure option +2 at start token number 0 or when choosing sure option -1 at start token number 3). The choice function computed from the gain trials is indicated in green and the one computed from the loss trials is indicated in red. For each monkey, three corresponding choice functions are shown for two identical sets of gamble and sure pairs that are presented for three corresponding start token settings (indicated on top of the choice functions). **(d)** The choice functions allow us to estimate the subjective value (SV) of the corresponding gamble option in units of value associated with the paired sure options (here: 'expected end token number') using the model-free certainty equivalent method. The probability of the monkey to choose a gamble depends on the difference between its own value and the value of the alternative sure option. When the value of the sure option is small, monkeys are more likely to choose the gamble. As the sure option's value increases, monkeys increasingly choose the sure option. The choice function allows us to estimate the point when the probability of choosing either the gamble or the sure option are equal [$P(G) = 0.5$]. At this point, the subjective value of the two options must be equal, independent of the underlying utility functions that relate value to physical outcome. Therefore, the subjective value of the gamble is equivalent to the corresponding sure option value at this indifference point. This value is referred to as the certainty equivalent (CE)⁶⁰. The grey arrows show the two CE values, indicating that the SVs of the gamble options in gain and loss trials are different, even so their outcomes are the same in an absolute value framework.

Second, R1 suggested that the monkeys adjusted their choice preferences to maximize the probability to earn reward. To test this hypothesis, we constructed a number of

'reward maximization models'. In these models, we assumed monkeys chose the options that maximized their cumulative reward. As the reviewer suggested, we considered not only the probability of getting reward in the current trial, but also that in future trials. These models generated an optimal strategy to maximize reward rate (i.e., the number of rewards that were obtained for a given number of trials). We then compared these optimal strategies with the actual behavior of the monkeys.

In all models, we determined the set of risk attitudes that maximized reward rate. Because of the complexity of our task this optimization problem was not easy to solve analytically. We therefore opted for a simulation approach. In these simulations, we determined the choice on each trial by comparing the utilities of gamble and sure option. The utility of an option can be calculated in a variety of forms. The differences in the assumed utility representation gave rise to the different models.

- (1) alpha token model, which includes 6 free parameters:
6 alpha for each start token state ($U_{\text{option}} = X^{\alpha} * P$),
in which U_{option} is the utility of the option, X is the token number that can be gained or lose when choosing the option, P is the probability of the outcome (X) that can occur, and α_{1-6} are the parameters determine the utility of option when start token number equal to 0-5.

Model #1 represents the hypothesis of R1 in its pure form. Here differences in risk attitude are purely local adaptations to the distance to the target (i.e., water reward), indicated by the current token number with a power form (i.e. X^{α} start token number)

- (2) alpha GL token model, which includes 12 free parameters:
6 alpha for each start token state in gain, 6 alpha for each start token state in loss.
- (3) alpha lambda model, which includes 12 free parameters:
6 alpha for each start token state (in both gain and loss),
6 lambda for each start token state if it is in loss ($-\lambda * (-X)^{\alpha}$).

Models #2,3 both presume that risk attitude is both affected by token number AND by the difference between potentially gaining or losing tokens. Note that these models assume a relative encoding of value, and therefore are conceptually closer to Prospect Theory.

- (4) alpha theta model, which includes 2 free parameters:
1 alpha for the general utility curvature,
1 theta to modulate the alpha based on start token number ($\alpha' = \alpha + \theta * \text{tkn}$).

Model #4 is a simplified version of model #1, where risk attitude shifts are presumed to be linearly related to token number.

- (5) alpha lambda theta model, which includes 3 free parameters:
1 alpha for the general utility curvature, 1 lambda for general loss utility modulation, 1 theta to modulate the alpha based on start token number.

Model #5 is a simplified version of model #3, where risk attitude shifts are presumed to be linearly related to token number. This model is closest to a conceptual framework whereby behavior is guided by strategic adjustments in risk attitude within a value encoding framework given by Prospect Theory.

After making a choice based on the utility comparison between gamble and sure option (using a choice function with inverse temperature as a free parameter between [0, 1]), we determined the choice outcome based on the winning/losing probability of the chosen option and then update the start token for the next trial. We repeated this process for ~10,000 training trials (one half of all completed choice trials of each monkey; 11,662 and 9,966 for monkey G and O, respectively) with a specific gamble-sure option set identical to the one used in our experiment.

Next, we found trials with specific start token (Tkn = 0~5). For each trial with a given start token number, we calculated the cumulative reward for the current trials by calculating the 'value of reward' for the following 10 trials (containing the current trials) with a specific discount rate K. For example, if the matrix of the subject to get reward for the current trials is [1 0 0 1 0 1 0 0], the cumulative reward will be calculated by $1+k^1(0)+K^2(0) + K^3(1) + K^4(0)+K^5(1) + \dots$. Then we average the cumulative reward for all trials with the same start token number. We repeated this process for the 6 different start token numbers.

For each of the 5 'dynamic' models, we found the free parameters that maximized the cumulative reward rate. For comparison, we also included the two 'static' models described in the previous manuscript: prospect theory and EU fitted separately for each start token number. In addition, all models had two more free parameter, discount rate K and the inverse temperature of the choice function.

We compared the models using two criteria. First, we computed the proportion of choices that the model correctly predicted for the testing trials (i.e., the other half of completed trial of each monkey). Second, we computed the AIC and BIC combining likelihood estimation with punishment terms for larger number of free variables. The results are summarized in the following table.

Dynamic models

cumulative trial number = 10

Model	Based economic model	free parameters	choice prediction	-LL	AIC	BIC
alpha_token	EU	6+2	0.78, 0.71	10651±4617, 12571±4751	21424±9536, 31908±11795	21483±9536, 31965±11795
alpha_GL_token	PT	12+2	0.77, 0.72	10936±2996, 11565±6244	21927±5882, 27327±14956	22030±5882, 27428±14956
alpha_lambda	PT	12+2	0.81, 0.71	18092±7780, 14870±5511	37021±15676, 40240±12140	37124±15676, 40341±12140
alpha_theta	EU	2+2	0.78, 0.71	14524±10122, 29017±3034	28507±20332, 67904±8040	28536±20332, 67933±8040
alpha_lambda_theta	PT	3+2	0.81, 0.72	16920±3261, 11384±7363	34732±6965, 30712±18180	34768±6965, 30748±18180

Static models

Model	How start token number influence risk-attitude/choice	free parameters	choice prediction	-LL	AIC	BIC
PT in manuscript	PT	30	0.83, 0.78	6431, 6926	12923, 13913	13164, 14150
EV in manuscript	-	12	0.82, 0.78	7344, 8212	14713, 16449	14808, 16544

Table 1. Model comparison.

EU, expected utility model that uses the start token number to influence the utility curvature; PT, prospect theory model that also uses the start token number to influence the utility curvature but takes the context of gain and loss into account; Free parameters +2 for dynamic models indicate the reward discount rate K and inverse temperature of the choice function; -LL, negative log-

likelihood for the model with parameters that maximize cumulative reward in 10 trials; AIC, Akaike information Criterion (computed with -LL); BIC, Bayesian Information Criterion (computed with -LL). The table for dynamic models summarizes for each model its fitting performances (choice prediction, -LL, AIC, and BIC) for each monkey with parameters that maximize the cumulative reward rate. The table for static models summarizes for each model its fitting performances for each monkey with parameters that maximize the likelihood estimation. The result of choice prediction, -LL, AIC, and BIC are shown for monkey G and O, respectively. Note: we ran all dynamic models for 5 times (each with at least 100 iterations) since the results of these models were not stable (see the standard deviation) as the results of the static models.

The main result from our modeling is as follows: (1) Both 'static' models fit much better to the monkey's behavior than any of the 'dynamic' models. This is true, even though they have many more degrees of freedom. In particular, the Prospect Theory model (fitted separately to all start token numbers) provides the best fit of all models. This seems to indicate that the hypothesis of reward maximization across trials is not the best explanation of behavior. (2) No 'dynamic' model fits behavior best consistently for all criteria and monkeys. According to choice prediction as the criterion, the 'alpha_lambda_theta' model fits behavior best for both monkeys. According to AIC/BIC as the criterion, the 'alpha_token' model works best for monkey G, while the 'alpha_lambda_theta' model works best for monkey O. However, the model fits overall are quite variable and the differences between the best-fitting and second-best models often are very small. Altogether, the 'alpha_lambda_theta' model seems to be the best model across monkeys and criteria. This model presumes a difference between gains and losses, but also includes an adjustment of risk preference as a function of token number. This finding fits with the best-fitting static model, which has the same characteristics.

Third, we also tested if anterior insular cortex (AIC) neurons encoded value in a relative framework (representing gains and losses of token) or in an absolute framework (representing the expected end token number). The results are again clear, AIC neurons use predominantly a relative value framework.

In conclusion, we thank R1 for suggesting the possibility that the start-token-dependent shift in risk attitude we observed could be due to dynamic statistical programming. It is a valid hypothesis, yet our findings do not robustly support it in its strict form. We have found behavioral and neural evidence that monkeys use a relative value encoding scheme, similar to the one proposed by prospect theory. The reference point for this relative value framework is the currently owned token number. We have neuronal evidence that this reference point is encoded in AIC as well. Thus, the behavior and neuronal activity in AIC is well explained by prospect theory.

These findings cannot be explained by stochastic dynamic programming alone, because the monkeys do not behave as if they exclusively maximize reward rate without taking the context of gain and loss into account. Nevertheless, we do not think that dynamic programming and Prospect theory are mutually exclusive. The reward maximization models that showed the best fit with behavior assumed a basic prospect theory representation of value that was dynamically modified by token number. This shift could

be the result of strategic planning over multiple future trials, as suggested by R1, but the evidence is somewhat inconclusive. We spend a lot of work on developing and testing the various dynamic reward maximization models. However, we think that the results of this effort are not clear enough to be reported in this manuscript. The static models provide a much better fit to behavior than any of the dynamic models. That implies that even if the monkeys were able of strategic planning, they did not use it consistently. Furthermore, the outcome of the behavioral fits was very variable and none of the dynamic models was consistently better than all the others across evaluation criteria and monkeys. Thus, in the end we decided not to include the modeling results in the paper. However, if R1 disagrees, we can include the results of the reward maximization modeling into the paper.

Contrary to the claims of the authors, the observed behavior has NOTHING to do with prospect theory. Prospect theory is relevant to situations where a reference point appears to be used when expected utility theory prescribes that no reference point should be used. In Kahneman and Tversky's famous experiment (Econometrica, 1979), IDENTICAL gambles ("prospects") are proposed, but framed differently. The frames make subjects think in terms of an imaginary reference point, and losses and gains. Behavior changes when the frame, and hence, the reference point, is manipulated. In contrast, in the task presented here the reference point is not imaginary. It is REAL. The "frame" is a target that moves depending on how many points have already been accumulated.

We agree with R1 that it is misleading to refer to the preference changes as evidence for 'framing', because the choice trials are not identical in all aspects. R3 had a similar concern. In the revised manuscript, we have removed the reference to framing. Instead, we describe the results as a test of the absolute versus relative value framework, as described (**Fig. 1e**, **Suppl. Fig. 4**, lines 131-148).

However, we disagree with R1 that framing (in the sense of using verbal instructions to direct the attention of the decision maker to specific aspects of the choice situation) is the only or most important aspect of Prospect Theory. Prospect Theory rests on a number of fundamental assumptions: (1) utility is represented in a relative reference frame, with respect to a reference point that can shift depending on the context of the decision, (2) utility functions in the gain and loss domain are different from each other (reflection effect), (3) utility function in the loss domain are steeper than in the gain domain (loss aversion), (4) the probability estimates used for decision making are a non-linear weighted function of the objective probabilities.

Our current findings in anterior insular cortex (AIC) do not allow any conclusions regarding the mechanisms underlying the non-linear probability weighting functions. However, the monkey's choice behavior and AIC findings indicate that all of the other 3 assumptions reflect real neuronal mechanisms used in the primate brain and are not simply abstract mathematical functions that efficiently predict behavior. This finding is the essential core of our manuscript.

The specific setup was taken from Seo and Lee, Journal of Neuroscience, 2009. There is, however, an important difference. Unlike here and in Symmonds et al (2010), uncertainty was resolved through play in a biased game of matching pennies. The monkey played against a computer that changed its strategy to exploit non-Nash behavior. Importantly, unlike here and in Symmonds et al (2010), pure strategies (always playing the SAME given the points accumulated and gambles presented) are NEVER optimal. As an aside, Seo and Lee (2009) also report that their subjects behaved nearly optimally. This is quite remarkable since Seo and Lee assume risk neutrality. Presumably, learning (to play the right strategy) dominates any effect of risk aversion.

As we point out in the manuscript, we took the idea for the token design from the paper by Seo and Lee (2009). This was important because it allowed us to expose the monkeys to losses. However, as R1 correctly notes, our token gambling task and the matching pennies task used by Seo and Lee vary in many other aspects. They have completely different purposes and different designs. One of the most important ones is the fact that our task represents a 'game against nature' (task contingencies are uninfluenced by the decision-makers choices), while the matching pennies task is a 'game against an agent' (outcomes depend both on the decision makers' choices and the ones of the other agent). Because of this we do not discuss the Seo and Lee (2009) paper further in our manuscript. Instead, our paper is conceptually more related to the design in two other studies: (1) Juechems, K., Balaguer, J., Ruz, M. & Summerfield, C. Ventromedial prefrontal cortex encodes a latent estimate of cumulative reward. *Neuron* 93, 705-714. e4 (2017).; (2) Farashahi, S., Azab, H., Hayden, B. & Soltani, A. On the flexibility of basic risk attitudes in monkeys. *J. Neurosci.* 38, 4383–4398 (2018). Neither of these studies mention 'dynamic programming' or learning but focus on how subjects' risk-attitudes were influenced by different contextual factors.

Incidentally, in the analysis of the experiment on which the present one builds, Seo and Lee (2009), there is NEVER any mention of prospect theory. This is as it should be. Seo and Lee do analyze the difference in behavioral and neural effects after gains and losses, but this is with a focus on LEARNING (to play the right strategy). By then, it was well known that animal learning in gain and loss conditions differ. Prospect Theory does not cover learning. Prospect Theory is about choices in stable conditions, when the probabilities of the outcomes are (supposed to be) known. Among others, it is well known that the usual inverted-S shaped probability weighting from Prospect Theory (confirmed here and in Stauffer et al, 2015) flips to an S-shaped function under learning (see the discussion in Hertwig and Erev, TICS 2009).

The recording sessions were only started after extensive training of the monkeys for >12 month. At this point their behavior was stable across sessions and showed regular dependencies on the cues used in the task design (i.e., color = token number gain/loss; relative area = probability of outcome; number of filled circles = currently owned tokens). Thus, we are confident that the monkeys had learned the task conditions, including the contingencies for receiving reward, and the meaning of the visual cues.

The only semblance of prospect theory in the monkeys' behavior is overweighing of small probabilities. This has been observed for monkeys before: Stauffer et al., Journal of Neuroscience 2015 (Reference 26 in the paper). This in itself, however, does not violate expected utility axioms. Agents are allowed to employ subjective beliefs, as long as they do not violate principles of rational choice such as the sure thing principle. Nothing in the reported results suggests that the monkeys made irrational choices.

We presume that R1 would consider any deviation from expected utility theory as irrational. As we have outlined above, we found strong behavioral and neuronal evidence that the monkeys evaluate options as gains or losses relative to a reference point. The use of such a relative value framework is clearly not in accordance with expected utility theory that presumes an absolute value framework. In that sense, we would suggest that the monkeys indeed made irrational choices.

However, this means of course not that the monkeys did not choose optimally given the relative value representation and the higher sensitivity to loss. Furthermore, it does not rule out the possibility that the monkeys strategically adjusted the relative value signals depending on how close they are to receive reward.

I found the imaging results most interesting, and very well presented.

We thank the reviewer for this positive assessment.

I would urge the authors however to refrain from presenting their case as a study of prospect theory, and follow prior literature in correctly interpreting their task as one of stochastic dynamic programming rather than prospect theory. I will not be able to formulate a definite recommendation about this paper until the authors do so. An interpretation based on prospect theory is misleading. It is analogous to interpreting outer space images with false colors where the false colors are not even inspired by position on the spectrum.

Clearly, the main difference between our interpretation and the one favored by R1 is conceptual. R1 argues that prospect theory is not the correct conceptual framework for interpreting our data. Instead, R1 favors statistical sequential programming (i.e., the adjustment of the value of options as a function of how close one is to receiving a reward). In response, we would argue that (1) Dynamic programming in its strong form implies the use of an absolute reward framework, because such a framework naturally captures how close the monkey is to receiving a reward. However, there is very strong behavioral and neuronal evidence that monkeys use a relative framework, as implied by Prospect theory. That means that dynamic programming cannot explain the most relevant aspect of the monkeys' behavior, the different risk attitude for gains and losses. (2) Dynamic programming (in a weaker form) and Prospect theory need not necessarily be mutually exclusive. However, while dynamic programming can potentially explain one aspect of the monkeys' behavior (the change in risk attitude as a function of start token

number), the evidence for reward maximization as a guiding principle of monkey behavior is not conclusive.

Thus, we conclude that our behavioral findings still support the use of Prospect theory that assumes a relative value framework. Such a framework requires a reference point and the representation of option values as gains or losses. We found such signals in the AIC. We think this is not a trivial finding, exactly because an absolute value framework, in which the brain registers how close it is to receiving reward within a sequence of actions, is a completely reasonable framework for encoding value and for making decisions. Thus, the fact that we found almost no evidence for value signals using an absolute framework, very strongly supports our conclusion that AIC encodes value in a framework similar to prospect theory. Thus, our findings provide evidence that prospect theory is not just a successful descriptive model of behavior, but instead captures critical aspects of the actual decision system implemented in the brain.

Reviewer #2 (Remarks to the Author)

We thank Reviewer #2 (R2) for his positive assessment of our work and the positive and constructive feedback.

Abstract:

- *the token task is not all that novel and was used by Lee before (2009 JN). This is mentioned later, but in the Abstract the word 'new' should be removed.*
- *what the authors label as 'asymmetric value function' is correct but is more commonly and specifically referred to as 'loss aversion' (steeper loss than gain function, mentioned in Results section with fig 1). The authors may consider to rather use the term of loss aversion that is more specific than 'asymmetric value function'.*

Thank you for the comments. In the revised manuscript, we have modified the abstract (line 28 and 30-31) as suggested.

Fig 1b:

- *quotation of this fig is reversed in the Results text: gain is actually shown at left and loss at right. (I would prefer to start the description of the choice options with gains for convenient relation to previously mostly studied gains, so the sequence would be fine as in the text but should be reversed in fig 1b.)*

We reversed the position of gains and losses in the figure as suggested.

- *it is unclear what exactly were the two options the animal was presented with: the text says 'choose between a certain token increase versus an uncertain option that could result in an even larger increase or no increase', and fig 1b does not contain that information either. And by 'increase versus even larger increase or no increase', do you mean a large safe reward vs. a gamble with one larger and one smaller reward than the safe option'? Please clarify. Correspondingly for losses. And please add this information into fig 1b.*

We changed the description of the option set that was presented to the monkeys. In the revised manuscript, we changed the main text (lines 90-95): "Thus, in the gain context, the monkey had to choose between a sure option that resulted in a certain token increase, whose size varied across trials, versus a gamble option that could result in a large increase or no increase at all, with varying outcome probabilities across trials (**Figure 1b**, left). In the loss context, the monkey had to choose between a certain loss and an uncertain option that could result in no loss at all or a large loss (**Figure 1b**, right)." We also changed the relevant figure legend (**Figure 1b**) (lines 892-903):

"(b) Set of reward options. Each option was represented by a square cue. Sure options had one color, while gamble options had two colors, indicating the two possible outcomes. The colors indicated the possible outcomes (ranging from -3 to +3, in units of token change). For gamble options, the portion of colored area indicated the probability of the corresponding outcome to be realized ($p = [0.1,0.9], [0.5,0.5], [0.75,0.25]$). The trials were divided into a gain and a loss condition because the possible outcomes of a given trial were either all gains or all losses of tokens or no token change. In choice trials, there was always a sure and a gamble option offered. Three gamble options could result

in a gain, while three other gambles could result in a loss. Each gamble option was paired against all sure options ranging from best to worst possible gamble outcome, resulting in 24 possible combinations (half gain and half loss context). In forced-choice trials, only one option was presented, which was one of the 13 different reward options (7 sure and 6 gamble options) which were used in the choice trials. See Methods for details.”

We hope this description is clearer. Please see also the task description in the Methods section (lines 435-497), in particular the last section lines 477-497.

• please label probabilities in the legend as $p = [0, 1]$, not in %, as per general convention.

We changed the label for the probabilities throughout the revised manuscript.

Fig 1f: what the authors call subjective value (SV) is usually called Expected Utility in standard economic notation [EU = SUM (u(x) • pi(p))]. Please stay with accepted terms for clarity throughout the paper.

Thank you for pointing out these problems. We have modified **Figure 1** accordingly.

Risk attitude (fig 1). These are very nice tests and fabulous results. It is a bit surprising that the animals were generally more risk seeking with gains than with losses, but the important thing is that they show loss aversion (steeper loss than gain function; at least monkey G) and that risk attitude changed with increasing gains towards aversion (concave utility). It is also reassuring to see that their gain utility function became less convex and more concave with higher start amounts (wealth, fig 1g top).

Thank you for the positive feedback. This is a nice summary of our main findings.

Line 143: the diction ‘while subjects exhibit diminishing marginal sensitivity’ sounds a bit odd: concave utility indicating risk aversion IS DUE TO diminishing marginal utility (not ‘sensitivity’, unless you want to diverge into Weber’s law). One could just say ‘due to diminishing marginal utility’.

We have modified the text (line 157) accordingly.

Probability weighting functions: please cite the specific weighting function you are using (Prelec, Gonzalez-Wu or Kahneman-Tversky) to link the reader to proven methods. You may also consider citing one or both behavioral probability weighting studies by Stauffer et al. 2015 and/or Ferrari-Toniolo et al. 2019 (both J Neurosci) that present more details and support your behavioral analysis and conclusions.

In the revised manuscript, we describe the weighting function we are using: (lines 172-176): “Objective probabilities are mapped onto a subjective weighting function as follows: $w(p) = p^\gamma / (p^\gamma + (1-p)^\gamma)^{1/\gamma}$ ^{2,6,21}. $\gamma > 1$ indicates an S-shape subjective probability mapping (overestimated for large probabilities and underestimated for small probabilities), $\gamma < 1$ indicates an inverse S-shape subjective probability mapping (underestimated for large probabilities and overestimated for small probabilities), and $\gamma = 1$ indicate a linear mapping of objective probabilities.”

This function is the one suggested by Tversky & Kahneman (1979) [#6 of our citations]. We also cite Juechems et al., (2017) [#2], which uses the same function and describes code for fitting. In addition, we cite Stauffer et al., (2015) [# 21] (line 178) as a study that supports our findings.

Paragraph starting on line 180 and fig S3: a comparison between a linear EV model and a nonlinear utility model should, in such a sophisticated study as presently, be done via choices rather than the less sensitive reaction times: would the choices be better explained by EV or utility (with or without probability weighting)? Using reaction times for such a comparison is interesting but a step down from standard-of-the-field fits to choices.

We are sorry for the misunderstanding. In the previous manuscript, we referred to the wrong Suppl. Figure. Thank you for pointing out our mistake. In the revised manuscript, our conclusion is now supported by **Table 1** showing the results of the model comparison for the EV and the Prospect theory (PT) model. The comparison between the best fit of the EV and the PT model to behavior is shown in **Suppl. Fig. 5**.

Line 181: utility IS subjective by definition: ‘subjective utility’ might sound odd as it implies that there is an objective utility, which does not exist by definition.

We changed the wording throughout the revised manuscript (‘subjective utility’ to ‘utility’ and ‘subjective value (SV)’ to ‘expected utility (EU)’).

Neuronal data section starting on line 185: the variety of neuronal modulation types is bewildering (eg. Lines 195-203, which are difficult to understand for me), and the description is a bit scattered across several paragraphs and intermingled with anatomical location. It is also unclear how many neurons were tested in no-choice trials and how many in choice trials. I think it would help to make an initial overview with clear definitions, descriptions and numbers of each modulation type, also referring to Table 1 (which is appropriately labelled as ‘model comparison’ but does not inform about the character of the many modulation types) and Figure 2. Maybe make a succinct paragraph for each neuron type, with definitions and numbers of neurons from Table 1 and reference to the relevant parts of fig. 2, with longer paragraphs for the more numerous or more important neuron types. Then go into details of the gain-loss sensitive neurons (fig 3).

Thank you for the feedback. As suggested, in the revised manuscript we now start the section describing our neuronal results with an initial overview of the different types of signals we have found (lines 215-231). Afterwards we describe the different signal types in detail in order of frequency of occurrence (lines 232-272).

Figure 2: it would be reassuring to have a raster plot for at least one modulation type.

In the revised manuscript, raster plots for each of the example neurons in Fig. 2 are shown in **Suppl. Fig. 7**.

And replace 'linear value' in the title of part c by 'monotonic value' or something similar, to avoid the confusion stated in my next paragraph.

In the previous manuscript, we used 'linear' simply because a linear regression fitted the data well. We did not want to make any particular claims that the relationship between neuronal activity and any of the behavioral variables was necessarily strictly linear. Given this possible misunderstanding associated with our terminology, we decided to avoid the term 'linear' in general. We have changed the terminology of several signals. We changed 'linear value' to 'general value' (to emphasize the difference to the context-dependent gain and loss value signals); 'linear token' to 'parametric token'; and 'linear risk' to 'parametric token'. We made changes in the main text (line 234, 247, 264), methods (line 638, 648, 662), figures & figure legends (**Figure 2, Suppl. Figure 7**), and tables (**Table 2, Suppl. Figures 1,2**) accordingly.

Line 211: in a paper in which utility functions are fitted to choices, the term of 'AIC neurons reflected information about the expected value of the options' can have two meanings: (1) if the neurons are suggested to code expected value ($x \cdot p$) as opposed to expected utility ($u(x) \cdot p$, or $u(x) \cdot pi(p)$), one would need to show a significantly better regression fit with expected value than expected utility. I don't see these data in the paper, nor an analysis that would make this distinction. (2) if the neurons are simply increasing their activity monotonically, a linear regression could be significant, as could be a rank correlation (eg. Spearman), but the result should not be interpreted as linear coding as opposed to non-linear coding if the comparison has not been made. It seems to me that the latter was the case, as line 213 correctly says 'monotonically rising'.

Lack of analysis for expected utility coding vs. expected value coding would be fine given the complexity of the analyses already, but the language is important.

As indicated above, we did not intend to make a strong claim that the AIC neurons encoded EV, but not EU. R2 is correct in presuming that we only want to claim monotonic relationships. Given the noisiness of the spiking activity, we did not record enough trials and enough different value samples to be able to clearly differentiate these two types of signals. In the revised manuscript we now state: (lines 246-251). "The second largest group of AIC neurons reflected information about the value of the options (49%; 73/149). A subset of this group of AIC neurons (19%; 14/73), carrying a 'General value' signal (**Figure 2c**), encoded the option values in a monotonically rising (18%; 13/73) or falling (1%; 1/73) fashion, for both gains and losses. Such neurons could either encode expected value or expected utility, but our data were not sufficient to distinguish between these possibilities. This kind of value signal is not gain/loss context sensitive."

Paragraph starting on line 308: what does it mean when a neuron has a 'significant choice probability'? In the paragraph above, ROC AUC discriminability is used as definition of choice probability. I appreciate this comes from ROC theory, but it is worth explaining the intuition behind this in a paper dealing with probability of empirically measured behavioral choice ($p(\text{choice})$). – If this is properly explained, the result seems that all types of neuronal modulation can predict either the choice or the risk attitude, which is a very nice result worth being more explicitly presented.

Thank you for the suggestion. In the revised manuscript, we avoid the term 'choice probability', because it is associated with perceptual decision making, as R2 correctly notes. Instead, we refer to the area under the curve (AUC) of the receiver operation characteristics (ROC). We briefly describe the general notion that AUC is a measure of the probability that an ideal observer can correctly distinguish between two different firing rate distributions of a neuron in two different states. Next, we introduce two different AUC measures. One measures the correlation of AIC activity fluctuations with two overtly distinct behavioral states (i.e., the choice of either a sure or a gamble option). This is the AUC of 'explicit choice'. The other measures the correlation of AIC activity with two internal states (i.e., high or low willingness to take risks). These 'internal states' are not associated with overt behavioral markers but can be identified by choices that do not maximize expected value. This is the AUC of 'implicit risk attitude'. We finish with the conclusion that (lines 346-349): "The AUC analysis showed that fluctuations in the activity of some AIC neurons are predictive of choice and/or risk-attitude. This suggests a causal role of AIC in risky decision-making. AIC neurons might encode value information, on which the monkeys' choice is based and contextual information that influences momentary risk-attitude."

Line 289: unclear what the term 'predict the internal state' contributes / means here?

In the revised manuscript, we changed the wording of this subtitle to (line 310): 'Choice- and Risk-attitude-related response modulation of AIC neurons'.

Anyway, congratulations for a nicely pioneering study with solid conceptual background rarely found in neurophysiology of this kind.

We appreciated the positive feedback very much.

Reviewer #3 (Remarks to the Author):

This manuscript investigated choice behavior in macaque monkeys when they choose under conditions where the potential outcomes are either gains and losses, and recorded responses from anterior insula cortex (AIC) neurons. The main results found that, similar to humans, monkeys weigh risk differently when faced with potential gains or losses, and that AIC neurons encode reference points for this gain/loss effect, as well as reference-dependent values, suggesting a role for AIC in decision-making under risk. Overall, I think this is a well-designed study with rigorous analyses. The results are important to the field because there is very little known about the neurophysiology of AIC. For instance, the finding that there is a relative prevalence of loss encoding among these neurons is important, as it is relatively uncommon to find valence selectivity at the level of cortical regions, and likely relates to findings on fMRI of insula activation in response to negative events/information.

We thank R3 for his positive assessment of our work and the positive and constructive feedback.

I only have a few comments regarding the results and interpretation. First, the authors note that their recording locations were near gustatory cortex, which is in the anterior insula. Although this study didn't set out to test gustatory responses, was there any evidence for this? For example, were there neurons that responded when the animal received the primary reward?

This is an interesting question. We did find a large number of AIC neurons (106 of 240, 44%) that were responsive, when the water reward was delivered. A few example neurons and their response dynamics are shown below. Many of these neurons exhibited also changes in activity when the outcome was revealed, which preceded water delivery by 900ms. Such anticipatory activity complicates the interpretation of responses to the water reward. Nevertheless, we found at least 77 (32%) neurons that specifically responded to the delivery of water reward.

Unfortunately, our experiment was not designed to study gustatory responses and we did not deliver reward with different taste. We are therefore not able to distinguish between neurons encoding specific gustatory stimuli (e.g., a particular kind of taste), and neurons encoding a more abstract reward signal.

We are currently working on a follow-up study, in which we analyze the AIC signals in the time period following the decision. We decided not to include the analyses of outcome-related activity in the present study, due to space limits and out of concern that it would make the manuscript unfocused.

Raster plots and spike density functions (SDFs) of example neurons respond to water reward delivery. Black dots/lines: trials with reward. Gray dots/lines: trials without reward.

Second, a difference between this task design and typical designs to test prospect theory in humans is that in humans, the same gamble can be framed as a gain or loss with language whereas here the gambles are actually for different outcomes. I think this distinction should be discussed in a little more depth, as well as any relevant studies from the human literature in which similar loss-only and gain-only contexts were tested and compared.

Yes, we agree that it may be misleading to present the results in **Fig. 1e** of the old manuscript as an evidence of ‘framing’, since the choice conditions on the gain and loss trials are not identical in all aspects (specifically, the start token number was different). R1 raised a similar concern as well. To avoid any possible misunderstanding, we have removed all references to ‘framing’ in the revised manuscript. Instead, we replaced the old results with a new analysis that tests, if the monkeys used an absolute or relative value framework.

We compare monkey’s risk attitude toward choice options that lead to identical outcomes (i.e., identical end token number), but resulted from either gaining or losing tokens. For example, a gain trial at start token number 0 with the gamble option [0, 50%; +3, 50%] versus the sure options [0, +1, +2, +3] has the same probability distribution of expected end token numbers as a loss trial at start token number 3 with the gamble option [-3, 50%; 0, 50%] versus the sure options [-3, -2, -1, 0] (**Suppl. Figure 4a**; see figure below). Similar pairings can be found for two other combinations of start token numbers and gamble options leading to gains and losses (**Suppl. Figure 4b,c**). We can determine the subjective value (SV) of the gain/loss gamble options in units of end token number by estimating the choice function for the gain and loss trials and determining the certainty equivalent of each gamble option. Because the outcomes are identical within an absolute framework (i.e., the expected end token numbers are the same), there should not be any substantial difference in subjective value, if the monkey uses an absolute value framework. However, we found that the SV of gamble options leading to gains were significantly higher than the SV of gamble options leading to losses (**Figure 1e**; **Suppl. Fig. 4d**; permutation test; monkey G & O: $p < 10^{-4}$ for gamble options that

resulted with the same end token number from either gaining or losing tokens). This result strongly suggests that monkeys evaluate options using a relative value framework.

These new results are described in the revised manuscript (lines 131-148) and shown in Fig. 1e and Supplementary Fig. 4 (shown below). Please see also our response to R1.

Supplementary Figure 4. Monkeys' risk preferences changed across gain/loss contexts with different start token numbers cannot be attributed to their evaluation of each available option based on the expected final token number

Prospect theory implies the use of a 'relative value' framework, where the value of an outcome depends on the change in assets relative to a reference point. Alternatively, the monkeys could use an 'absolute value' framework, where the value of an outcome depends on the final asset number. To test, which of these value frameworks is used by the monkeys, we compared trials with the same outcome in terms of final token number, but which resulted from either gaining or losing tokens, because the start token number was different. (a) For example, consider a trial with a start token number of 0, in which a gamble option with an equal probability of gaining 3 or 0 tokens is offered versus a sure option of gaining 2 tokens. The expected end token outcomes of this trial (owning 3 or 0 tokens, each with $p=0.5$ versus owning 2 tokens with $p=1$) are identical to a trial with a start token number of 3, in which a gamble option with an equal probability of losing 3 or 0 tokens is offered versus a sure option of losing 1 token. (b) The same two gambles can be systematically matched with other pairs of sure options that reached the same final token number by either gaining or losing tokens. The dotted boxes shows the probability distribution of expected end token numbers (i.e., the possible final token numbers and their probability) for each of the gambles and a set of corresponding sure options. The probability distribution of expected end token number for gain trials at a start token number of 0 (green dotted box) is identical to the one for loss trials at a start token number of 3 (red dotted box). Thus, the subjective value of the two gamble options should be identical in an absolute value framework, but different in a relative value framework. (c) Choice functions indicate the probability of choosing the gamble option as a function of expected end token number for the paired sure options. The expected end token number is a function of the indicated change in number of token and the start token number (e.g., expected end token number is 2 when choosing sure option +2 at start token number 0 or when choosing sure option -1 at start token number 3). The choice function computed from the gain trials is indicated in green and the one computed from the loss trials is indicated in red. For each monkey, three corresponding choice functions are shown for two identical sets of gamble and sure pairs that are presented for three corresponding start token settings (indicated on top of the choice functions). (d) The choice functions allow us to estimate the subjective value (SV) of the corresponding gamble option in units of value associated with the paired sure options (here: 'expected end token number') using the model-free certainty equivalent method. The probability of the monkey to choose a gamble depends on the difference between its own value and the value of the alternative sure option. When the value of the sure option is small, monkeys are more likely to choose the gamble. As the sure option's value increases, monkeys increasingly choose the sure option. The choice function allows us to estimate the point when the probability of choosing either the gamble or the sure option are equal [$P(G) = 0.5$]. At this point, the subjective value of the two options must be equal, independent of the underlying utility functions that relate value to physical outcome. Therefore, the subjective value of the gamble is equivalent to the corresponding sure option value at this indifference point. This value is referred to as the certainty equivalent (CE)⁶⁰. The grey arrows show the two CE values, indicating that the SVs of the gamble options in gain and loss trials are different, even so their outcomes are the same in an absolute value framework.

REVIEWER COMMENTS

Reviewer #1 (Remarks to the Author):

I have now studied the authors' rebuttal, and re-read the paper. In my final evaluation, I will be constructive.

One way to make my position clear is by pointing out that there is an important sentence missing in the Abstract. After the sentence "[t]hese behavioral effects are well described by Prospect Theory, with the key assumption that humans represent the value of each available option asymmetrically as gain or loss relative to a reference point," the following has to be made clear:

"Using a token gambling task where monkeys were rewarded only after reaching a minimal number of tokens across multiple trials, we induced changes in the reference points which allowed us to observe differential behavior in the gain or loss domain, and to explore the underlying neuronal computations."

In the analysis of the behavioral and neuronal data, the authors chose to ignore the dynamic structure of the task. If they had done so, the behavioral fit would have been worse, as the authors show in their rebuttal. The cost of ignoring the dynamic structure is, of course, complexity. The behavioral model has 30 parameters whereas the best alternative dynamic model has only 5. The neuronal categorization is correspondingly complex: the authors categorize based on 163 models, with consequent risk of mis-categorization and lack of power in subsequent statistical inference. In short, the consequence of preferring a model that fits behavior better statistically is a tremendous increase in complexity of the analysis of neuronal data.

Despite this, the results are informative, documenting how neurons in the AIC are encoding various aspects of the behavioral model (Figs 2, 3), even if relatively few neurons ultimately predict behavior (Fig . 4). From a neuroscience perspective, the job is done, very well in fact.

My critique of this work remains though. It would be far more informative if the authors had spent time on fitting a more parsimonious but more appropriate model to the behavioral data. This is obviously very challenging, since we don't know how exactly monkeys approach the complex stochastic dynamic programming that comes with the task, and because we know from observation of human choices in a similar task (Symmonds et al., Journal of Neuroscience) that humans deviate from fully optimizing.

I am missing discussion of these issues in the main body of the paper. I'm sure I will not be the only reader who will wonder why the authors ignored the dynamic structure of the task.

From their rebuttal, I sense that the authors do not understand that, if a more parsimonious model does not fit as well according to purely statistical criteria, such as AIC, it may still have more value scientifically. Here, the more parsimonious model recognizes the dynamic nature of the task, while the less parsimonious model ignores it, yet monkey behavior exhibits tantalizing regularities as a function of the state variable (Number of tokens held; Fig 1d), in line with human findings (Symmonds et al.). Should I remind the authors of the fact that Newton's model of the movement of planets initially did not fit the data as well as the then extant -- but now discredited -- model, which was full of "patches" (called epicycles and the sort; they are now considered patches, but at the time they had a logic though)?

Reviewer #2 (Remarks to the Author):

Thank you for the careful and considerate revision. I have no further comments.

Reviewer #3 (Remarks to the Author):

I have read the authors' response and revised manuscript and found that they sufficiently

addressed the points I raised in review. The text revisions are helpful and the new analysis is a positive addition to the study. I have no further concerns.

Response to Reviewers

We thank all three reviewers for their thoughtful comments on our manuscript. Two reviewers indicated that we addressed their concerns sufficiently. Here we provide a point-by-point response to reviewer #1's comments along with a description of how the paper was revised accordingly. Our response is indicated in blue.

Reviewer #1 (Remarks to the Author):

I have now studied the authors' rebuttal, and re-read the paper. In my final evaluation, I will be constructive.

One way to make my position clear is by pointing out that there is an important sentence missing in the Abstract. After the sentence "[t]hese behavioral effects are well described by Prospect Theory, with the key assumption that humans represent the value of each available option asymmetrically as gain or loss relative to a reference point," the following has to be made clear: "Using a token gambling task where monkeys were rewarded only after reaching a minimal number of tokens across multiple trials, we induced changes in the reference points which allowed us to observe differential behavior in the gain or loss domain, and to explore the underlying neuronal computations."

We do not disagree with the observation of reviewer #1 (R1) that the sequential nature of the task introduced changing reference points. In fact, we used this property of the task to specifically test situations in which the outcomes of the choice were identical in an absolute sense (final token number), but which represented changes in value relative to the reference point that were opposite of each other (gain/loss of tokens). Thus, the behavior of the monkeys, that is their choices and risk preferences, showed a consistent dependency on the relative value of the outcome (i.e., whether it represented a gain or a loss relative to the reference point). In addition, as we report in our paper, the monkey's risk attitude was also influenced by the changing reference point itself (i.e., the start token number), especially in the gain domain. We capture this effect as a modulation of the utility and probability weight functions by the start token number. As R1 has pointed out, one possible interpretation of this effect is the influence of some strategic adjustment based on awareness of the start token number as a state variable that indicates how close the monkey is to receiving reward.

We do not wish to deny the possibility that this is indeed the case. However, a number of considerations complicate the interpretation of this behavioral finding. First, such a strategic adjustment would not have changed the basic difference in risk attitude for gains and losses, but only modulated it. Strategic adjustment of risk attitude using an absolute value framework aimed at minimizing the number of trials in between rewards cannot explain important characteristics of the monkeys' behavior. Thus, the change of risk attitude as a function of start token number is not a reason to believe that the monkeys did not use a relative value system. Second, we observed this effect robustly only in the gain domain in both monkeys. Third, we do not have clear evidence that the monkeys engage in rational estimation of future outcomes based on the current token state. As R1 acknowledges, even humans perform such strategic adjustments not optimally. Providing evidence for such strategic adjustments and estimating to what extent they

are the result of rational, forward-looking processes are not easy in monkeys. Fourth, at least in principle other explanations are possible, such as a general influence of wealth on risk attitude, independent of rational considerations about future rewards.

In this paper we therefore decided to concentrate on the aspect of the monkeys' behavior that seems to us the more important one and the one for which we have clear evidence. The monkeys' behavior represents a clear deviation from expected utility theory and is well described by a generalization of prospect theory that uses the reference point as a state-variable. In the revised manuscript, we now discuss the influence of token number on risk preference in more detail and mention that strategic adjustments could be the cause of these effects.

(lines 414-420) "The behavior of the monkeys did not only indicate that they use a relative value framework (**Figure 1e**), but also that risk-attitude was modulated by the token assets (**Figure 1d**). This asset effect might be due to deliberate strategic adjustments of risk attitude to optimize reward rate (Mkael Symmonds, Peter Bossaerts, Raymond J. Dolan A Behavioral and Neural Evaluation of Prospective Decision-Making under Risk. *Journal of Neuroscience* 27 October 2010, 30 (43) 14380-14389; DOI: 10.1523/JNEUROSCI.1459-10.2010). Alternatively, the asset effect might represent a general influence of wealth on risk attitude, independent of rational considerations about future rewards. We described this effect as a state-dependent modulation of the prospect theory model (**Figure 1g-i**) without a specific commitment to either hypothesis. Future work will be necessary to distinguish between these different possible mechanisms."

In the analysis of the behavioral and neuronal data, the authors chose to ignore the dynamic structure of the task. If they had done so, the behavioral fit would have been worse, as the authors show in their rebuttal. The cost of ignoring the dynamic structure is, of course, complexity. The behavioral model has 30 parameters whereas the best alternative dynamic model has only 5.

R1 correctly notes that the static models we presented in the manuscript have more free parameters than all the dynamic models we tested (see Dynamic and Static models listed in table 1 below). However, we also tested two additional simplified versions of the prospect theory model described in the manuscript. In both of these models, risk attitude shifts are presumed to be linearly related to token number. The difference between the models is simply that one of them includes 2 additional free parameters describing choice bias and the slope of the choice function.

(1) alpha lambda theta model without bias and slope, which includes 4 free parameters: 1 alpha for the general utility curvature, 1 lambda for general loss utility modulation, 1 gamma for subjective probability mapping, and 1 theta to modulate the alpha based on start token number.

(2) alpha lambda theta model with bias and slope, which includes 6 free parameters: 1 alpha for the general utility curvature, 1 lambda for general loss utility modulation, 1 gamma for subjective probability mapping, 1 theta to modulate the alpha based on start token number, and 1 bias and 1 slope (inverse temperature) to modulate the choice function.

These less complex and therefore more 'parsimonious' static models (4 & 6 free parameters) still outperform all dynamic models. Thus, the better fit of the PT models does not depend on higher complexity. We decided not to include these simplified PT models in our last response,

because the PT model used in the manuscript has fewer assumptions regarding how start tokens modulate monkeys' risk-attitude. For example, in the two simpler models included here, we assumed a linear modulation of the curvature of the utility function by token assets (i.e., alpha). However, other simpler models are also possible, in which token assets modulate other parameters in the PT model (i.e., lambda, gamma, bias, slope, or all possible combinations of these variables). Importantly, the PT model used in the manuscript also outperforms these simpler versions using the BIC comparison. We apologize if this caused some misunderstandings.

Dynamic models

cumulative trial number = 10

Model	Based economic model	free parameters	-LL	BIC
alpha_token	EU	6+2	10651±4617, 12571±4751	21483±9536, 31965±11795
alpha_GL_token	PT	12+2	10936±2996, 11565±6244	22030±5882, 27428±14956
alpha_lambda	PT	12+2	18092±7780, 14870±5511	37124±15676, 40341±12140
alpha_theta	EU	2+2	14524±10122, 29017±3034	28536±20332, 67933±8040
alpha_lambda_theta	PT	3+2	16920±3261, 11384±7363	34768±6965, 30748±18180

Static models

Model	How start token number influence risk-attitude/choice?	free parameters	-LL	BIC
PT in manuscript	PT	30	6431, 6926	13164, 14150
EV in manuscript	-	12	7344, 8212	14808, 16544

Parsimonious static models

Model	Based economic model	free parameters	-LL	BIC
alpha_lambda_theta	PT (without bias, slope)	4	8676, 9673	17392, 19386
alpha_lambda_theta	PT	6	6746, 7066	13552, 14190

Table 1. Model comparison.

The neuronal categorization is correspondingly complex: the authors categorize based on 163 models, with consequent risk of mis-categorization and lack of power in subsequent statistical inference. In short, the consequence of preferring a model that fits behavior better statistically is a tremendous increase in complexity of the analysis of neuronal data.

This is a misunderstanding. The 163 different regression models used for neuronal categorization result simply from the fact that neurons multiplex and can carry multiple signals. We need to test therefore all possible combinations of the basic set of variables that we consider. This is an aspect of neuronal representation that is unrelated to the parsimony (or lack thereof) of the set of behavioral variables that is tested.

My critique of this work remains though. It would be far more informative if the authors had spent time on fitting a more parsimonious but more appropriate model to the behavioral data. This is obviously very challenging, since we don't know how exactly monkeys approach the complex stochastic dynamic programming that comes with the task, and because we know from

observation of human choices in a similar task (Symmonds et al., Journal of Neuroscience) that humans deviate from fully optimizing.

We respectfully disagree. First and most importantly, stochastic dynamic programming (the alternative model preferred by R1) cannot explain the basic finding that the monkeys strongly differ in their preferences for choice options that result in the same end state but represent gains and losses. This fact is important because it goes beyond considerations of model fit and parsimony. The behavioral pattern that we observe in the choices of the monkey cannot be explained by a 'rational' model of decision making that only aims to maximize reward. However, this exact pattern is predicted by Prospect Theory. This seems to us clear evidence in favor of Prospect theory. At the very least, since the behavior of the monkeys follows key predictions of the Prospect Theory model, it seems to be an appropriate model to use in our research. We tested the alternative model suggested by R1, but it provided a less satisfactory explanation of choice behavior. Second, the behavioral model we use is potentially quite parsimonious, as we pointed out earlier. We decided to use a less parsimonious version, since it fit the data better and increased the descriptive power. Importantly, as we now state explicitly in the revised manuscript, in this model we also do not commit to a specific mechanism as an explanation for the behavioral effect. In general, we regard the prospect theory model as remarkable parsimonious, since it can describe a large amount of behavioral data in a large variety of different situations, using a very small number of parameters (see for example: Ruggeri et al., Nat. Hum. Behav., 2020).

Despite this, the results are informative, documenting how neurons in the AIC are encoding various aspects of the behavioral model (Figs 2, 3), even if relatively few neurons ultimately predict behavior (Fig . 4). From a neuroscience perspective, the job is done, very well in fact.

Thank you for the positive feedback.

I am missing discussion of these issues in the main body of the paper. I'm sure I will not be the only reader who will wonder why the authors ignored the dynamic structure of the task.

While we have described the sequential changes in risk attitude with different start token number, we agree with R1 that we have not discussed possible reasons for this effect. We added a brief discussion of the possible static or dynamic cognitive process that might lead to the behavioral effect:

(lines 414-420) "The behavior of the monkeys did not only indicate that they use a relative value framework (**Figure 1e**), but also that risk-attitude was modulated by the token assets (**Figure 1d**). This asset effect might be due to deliberate strategic adjustments of risk attitude to optimize reward rate (Mkael Symmonds, Peter Bossaerts, Raymond J. Dolan A Behavioral and Neural Evaluation of Prospective Decision-Making under Risk. Journal of Neuroscience 27 October 2010, 30 (43) 14380-14389; DOI: 10.1523/JNEUROSCI.1459-10.2010). Alternatively, the asset effect might represent a contextual bias related to wealth level. We described this effect as a state-dependent modulation of the prospect theory model (**Figure 1g-i**) without a specific commitment to either hypothesis. Future work will be necessary to distinguish between the different possible mechanisms."

From their rebuttal, I sense that the authors do not understand that, if a more parsimonious model does not fit as well according to purely statistical criteria, such as AIC, it may still have more value scientifically. Here, the more parsimonious model recognizes the dynamic nature of the task, while the less parsimonious model ignores it, yet monkey behavior exhibits tantalizing regularities as a function of the state variable (Number of tokens held; Fig 1d), in line with human findings (Symmonds o.c.). Should I remind the authors of the fact that Newton's model of the movement of planets initially did not fit the data as well as the then extant -- but now discredited -- model, which was full of "patches" (called epicycles and the sort; they are now considered patches, but at the time they had a logic though)?

As we explained in our preceding responses, we do not think that the hypothesis that the monkeys risk attitude is modulated by strategic adjustments is unreasonable (provided it is seen as an additional mechanism that modulates a basic relative value representation). However, as we also explain, this hypothesis is not fully supported by the data and alternative explanations are possible. In the revised manuscript, we added a section to the discussion to indicate this overall assessment. Relatively simple modulations of our basic task would be helpful to address the question. The monkeys could be tested in a task, in which they are rewarded according to the final token number at the end of each trial. If different start token numbers still modulate risk attitude, it seems less likely that this effect is due to strategic adjustments. We plan such experiments.

REVIEWERS' COMMENTS

Reviewer #1 (Remarks to the Author):

I will respectfully disagree with the authors on a number of things and sign off.

Reviewer #2 (Remarks to the Author):

I had been happy with the previous revised version, and my judgement has not changed since then despite the on-going discussions.